# THE PRIVACY-HALLUCINATION TRADEOFF IN DIFFERENTIALLY PRIVATE LANGUAGE MODELS

## ABSTRACT

While prior work has studied privacy tradeoffs with utility and fairness, the impact of privacy-preservation on factual consistency and hallucination in LLM outputs remains unexplored. Given that privacy-preservation is paramount in high-stakes domains like healthcare, the factual accuracy of these systems is critical. In this study, we uncover and investigate a *privacy-hallucination tradeoff* in differentially private language models. We show that while stricter DP guarantees do not distort knowledge acquired during standard pre-training, they hinder the model's ability to learn new factual associations when fine-tuned on previously unseen data, as a result of which the model tends to hallucinate incorrect or irrelevant information instead. We find that the proportion of factual texts generated drops by 17-24% when models are fine-tuned on the same data using DP ($\varepsilon = 8$), compared to the non-DP models, and on average, the factuality scores differ by at least 3-5%. This disparity is further pronounced when pre-training with DP, where we find a 43% drop in the number of factually consistent texts. Our findings underscore the need for more nuanced privacy-preserving interventions that offer rigorous privacy guarantees without compromising factual accuracy.

## 1 INTRODUCTION

The development and deployment of large language models (LLMs) in high-stakes setting requires non-negotiable standards for both privacy-preservation and factual accuracy. LLMs that are exposed to sensitive information during training are susceptible to reproducing it in subsequent interactions, resulting in privacy violations (Carlini et al., 2021; Chu et al., 2024; Meeus et al., 2024; Kandpal et al., 2024), even in non-adversarial settings (Aerni et al., 2025); this is legally and ethically unacceptable in high-stakes domains such as healthcare and law, which involve sensitive data. As simple anonymization offers insufficient protection (Staab et al., 2024; Xin et al., 2024; Pang et al., 2024), differential privacy (DP) (Dwork et al., 2006) has emerged as the gold-standard paradigm to provably mitigate such privacy risks in language models (Carlini et al., 2019; Li et al., 2022; Xu et al., 2023; Yan et al., 2024; Hu et al., 2024).

Alongside privacy issues, LLMs are known to *hallucinate*, i.e., to generate factually incorrect outputs (Wang et al., 2024; Jiang et al., 2024a; Das et al., 2022; Asgari et al., 2025).[1] This issue poses serious risks, particularly in high-stakes tasks such as generating patient discharge summaries (Chung et al., 2025). These concerns have motivated a body of research that aims to evaluate and improve factual correctness in LLM outputs (e.g. Ji et al., 2023; Li et al., 2024). While prior work has investigated privacy and factual accuracy independently, no work has investigated the interaction between them, despite the clear need to achieve both in high-stakes settings.

In this work, we empirically investigate the trade-off between these two critical properties, specifically focusing on the guiding question: does differentially private training increase factual hallucinations in models? Our work is motivated by the potentially conflicting conditions conducive to each property. Privacy-preserving strategies are inherently designed to counteract memorization (Miranda et al., 2025; Kassem et al., 2023; Hans et al., 2024), which may inhibit the acquisition and, thus, the output of factual information (Lu et al., 2024; Merullo et al., 2025). In particular, because example-level DP limits the influence of individual training examples on the model, one might expect a privately trained model to struggle with reproducing *rare facts* (i.e., those that appear only a few times in the fine-tuning data), while still capturing information that is repeated more frequently and is less likely to be privacy-sensitive. However, our findings indicate that, rather than only reducing the generation of rare facts, private training problematically leads to a general increase in hallucinations.

---

[1] We use "halluncination" to refer to LLM generation of false or misleading information presented as fact. Unless stated otherwise, we restrict ourselves to information not supported by or contradictory to the training (pre-training or fine-tuning) data.

**Goals and Questions.** Concretely, we study the interplay of DP training (using the DP-SGD algorithm for various privacy budgets) and hallucination in open-ended generation from language models. We consider potential hallucinations relative to facts found both in fine-tuning and pre-training datasets through the following research questions:

(RQ1) What impact does *DP fine-tuning* have on hallucinations concerning facts present in the *fine-tuning data*?

(RQ2) What impact does *DP fine-tuning* have on the hallucination (or forgetting) of factual information already learned by the model from the *pre-training data*?

(RQ3) What impact does *DP pre-training* have on hallucinations concerning facts present in the *pre-training data*?

**Main findings.** Through a detailed experimental study, using both automatic (Min et al., 2023) and human evaluations, we find the following:

- **RQ1**: A more stringent privacy budget (i.e. smaller $\varepsilon$ for DP) leads to consistent increase in hallucinations, as measured both by automatic and human evaluations (automated factuality scores decline by 3-5 points from non-DP to DP models). This highlights a privacy-hallucination tradeoff for fine-tuning. Further, we find that the DP models tend to repeatedly hallucinate the same incorrect claims, suggesting a systematic encoding of inaccurate facts rather than random errors.

- **RQ2**: We find no significant difference in the hallucination of pre-training facts with or without DP fine-tuning. That is, the noise injected during DP fine-tuning does not obfuscate knowledge encoded by models in pre-training.

- **RQ3**: The DP-pre-trained VaultGemma-1B model hallucinates significantly more than a similar non-private Gemma-3-1B model (factuality scores decline by 4-11 points). In some domains, its hallucination rate is comparable to a GPT-2 model that was never trained on these facts. Thus, the privacy-hallucination tradeoffs from pre-training are significantly worse than for fine-tuning (RQ1).

In summary, our results show a privacy-hallucination tradeoff in language models across both pre-training and fine-tuning settings. Our findings highlight the need for more nuanced privacy-preservation mechanisms that can protect data privacy without increasing the hallucination rate of the models. Upon publication, we will publicly release our code and data required to reproduce our research, and to facilitate future work in this direction.

## 2 RELATED WORK

**Privacy in LLMs.** This risk of privacy leakage by language models has inspired work on provable privacy-preserving strategies such as differential privacy (Li et al., 2022; Miranda et al., 2025), as well as heuristics such as knowledge unlearning to reduce the influence of sensitive data points on the model parameters (Jang et al., 2023; Zhang et al., 2024) and knowledge editing to locate and modify neurons containing private information (Wu et al., 2023; 2024).

Privacy-preserving methods typically (either directly or indirectly) involve changes to the model's parameters. For instance, DP training is typically accomplished through a modified version of SGD, which involves clipping and noising the gradient update (Abadi et al., 2016). Knowledge unlearning methods trace and remove an approximate estimate of the influence of a training point on the model's parameters. All of these approaches contribute to some degradation in model utility: DP limits the information the model learns, while knowledge unlearning and knowledge editing affects useful non-private information the model has encoded. We focus on the rigorous and future-proof guarantees of DP since heuristic privacy defenses can often be broken (Aerni et al., 2024; Du et al., 2024).

**Tradeoffs from DP.** The classical no-free-lunch theorem of DP states that DP necessarily incurs a penalty on utility (Kifer & Machanavajjhala, 2011). In practice, this results in a privacy-utility-compute tradeoff (McMahan et al., 2018; Ponomareva et al., 2023). Research has since established that privacy also comes at a cost to fairness in statistical estimation tasks (Tran et al., 2021), discriminative models (Bagdasaryan et al., 2019; Farrand et al., 2020; de Oliveira et al., 2024), and LLMs (Lyu et al., 2020; Matzken et al., 2023; Ramesh et al., 2024; Hansen et al., 2024). Ngong et al. (2025) explore the adverse effect of DP on elements such as grammatical correctness, fluency and the coherence of model-generated text. We note that it is possible to optimize for more balanced privacy-utility tradeoffs (Mireshghallah et al., 2021) or privacy-fairness tradeoffs (Pillutla et al., 2024; Zhou & Bassily, 2024).

Specifically in the text domain, prior work has empirically examined the privacy-utility tradeoffs with task-specific measures of utility, including classification accuracy, linguistic aspects such as fluency, grammatical correctness and lexical diversity. In contrast, we focus on tradeoffs between DP and hallucination. Minimizing hallucinations is highly desirable, and is often distinct from other task-specific measures of utility surveyed above.

**Factuality in Language Models.** No single factuality metric generalizes across settings, so it is common to use task- or domain-specific methods. Early research focused on measures such as the factual precision of cloze-style or short-

form responses (Youssef et al., 2023; Petroni et al., 2019), and NLI-based methods to determine whether generated summaries are consistent with their source document (Chen et al., 2021; Fabbri et al., 2022; Laban et al., 2022). The rise of LLM applications with open-domain, free-form model-generated text, where there can be multiple plausible responses has led to the development of factuality metrics in these settings (Min et al., 2023; Wei et al., 2024; Song et al., 2024). This is significantly more challenging, as the decoding strategy also influences model outputs (Wang et al., 2024), and models can generate correct answers across multiple attempts (Tian et al., 2023). Further, they may generate factually accurate content that contradicts or is unsupported by the training data (Cao et al., 2022).[2] We leverage existing state-of-the-art methods for fact-checking in open-ended generation along with human evaluations to analyze privacy-hallucination tradeoffs in LLMs (Min et al., 2023).

## 3 Experimental Design

Our goal is to measure how factual hallucinations are affected by DP at various privacy budgets. To this end, we carefully construct experimental setups where we can fine-tune valid differentially private models, control for overlap between fine-tuning and pre-training data, and evaluate factual accuracy in open-ended model outputs. As the specific choices of datasets, models, and training setups are crucial for achieving these requirements, we include those details within the broader description of our approach. Our experimental design is reusable for facilitating future analyses of LLM behavior, especially settings that similarly require careful separation of data not included in pre-training.

### 3.1 Models and Training Setup

**Fine-tuning setup for RQ1 and RQ2.** We fine-tune LLMs for controllable text generation, similar to previously proposed applications for DP in LLMs, particularly for privacy-preserving synthetic and open-ended text generation (Yue et al., 2023; Mattern et al., 2022; Ramesh et al., 2024). We provide a brief overview of the settings here and discuss the precise details and detailed background in Appendix A.

DP fine-tuning is achieved using a stochastic gradient optimization approach known as DP-SGD (Abadi et al., 2016). This algorithm bounds the information learned from each sample by clipping the per-sample gradients to a fixed $\ell_2$ norm bound, and perturbs them (for DP) with white Gaussian noise. The scale of the Gaussian noise is calibrated to the desired $(\varepsilon, \delta)$-DP guarantee.

The DP guarantees are provided with respect to the add-or-remove adjacency at the sequence-level, i.e., the model outputs should be nearly indistinguishable if a new sequence of 1024 tokens is added to or removed from the training dataset. We set privacy budgets of $\varepsilon \in \{8, 16\}$, and $\delta = n^{-1.1}$, where $n$ is the dataset size (in terms of number of sequences). We use the DP-SGD implementation from Opacus (Yousefpour et al., 2022) and measure the privacy budget consumed using the PLD accountant (with amplification by sampling) (Doroshenko et al., 2022). All fine-tuning (DP and non-DP) is achieved using Low-Rank Adaption (LoRA) for computational efficiency (Hu et al., 2022). We specify additional details about LoRA and all hyperparameter settings in Appendix B.

The base language model behind all fine-tuning experiments is GPT-J 6B (Wang & Komatsuzaki, 2021). This is a highly-performant decoder-only transformer model, similar to currently popular models. We make this choice for a crucial reason: its pre-training dataset, namely The Pile, is fully open and known, with a known cutoff date. This lets us ascertain that GPT-J is not pre-trained on any post-2020 data. This knowledge of the pre-training data mixture and cut-off date allows us to select fine-tuning data that has no overlap with pre-training data; cf. §3.2.

**Private pre-training for RQ3.** We address the effect of DP pre-training (as opposed to fine-tuning) on factuality using the recently released DP-pre-trained VaultGemma (VaultGemma Team, 2025). This is a 1B-parameter open-weights model fully pre-trained with DP ($\varepsilon = 2$); this is the largest model known to be pre-trained from scratch with DP as of this writing. We compare VaultGemma with the same two models as in VaultGemma Team (2025):

- Gemma3-1B, which can be viewed as the "non-private counterpart" of VaultGemma, and is expected to be similar except for the use of DP.

- GPT-2 XL (1.5B), whose performance is similar to VaultGemma on standard benchmarks. It has a much earlier knowledge cut-off date, as it was released in 2019.

---

[2]We regard this as hallucinations for the purpose of this work as the training dataset is considered as the sole *source of truth.*

| Dataset Name | Size | Examples/Description |
|---|---|---|
| Wikipedia Science | 231 | Follicular drug delivery; Malaria therapy; Eurotrac |
| Wikipedia AI | 124 | DeepSeek; DeepSeek (chatbot); DeepSeek (disambiguation); DALL-E |
| Wikipedia Pretraining | 250 | [randomly sampled articles] |
| Fine-tuning Data | 20355 | Wikipedia Science + Wikipedia AI + 20,000 randomly sampled Wikipedia articles |

Table 1: Datasets used for evaluation and fine-tuning.

## 3.2 DATASETS

Choosing a dataset for this study requires careful consideration of two factors. First, DP guarantees hinge on the assumption that the private fine-tuning data should not have appeared in the pre-training corpora of the LLM (Tramèr et al., 2024; Cummings et al., 2024). The importance of not violating this condition can be attributed to i) the potential for pre-training data to be adversarially extracted (Ishihara, 2023), and ii) evidence that pre-training and and fine-tuning on the same data artificially inflates performance estimates (Igamberdiev et al., 2022). Second, the inclusion of factually verifiable information and statements in the fine-tuning data is essential to evaluate changes in the factual correctness of the model's outputs; domains and datasets (e.g., social media posts) without clear factual content cannot be assessed for factuality.

In view of these two factors, we focus on Wikipedia data for fine-tuning and evaluation, where content is constructed to contain verifiable facts rather than opinions or speculation, and automated fact-checking methods have been previously validated (Min et al., 2023). Additionally, the articles' meta-data allows us to select articles created after 2020, ensuring they were not included in GPT-J 6B's pre-training corpora. We expect our fine-tuning data to have some overlap with text in the pre-training data in terms of linguistic patterns, broad concepts, and topics (in some settings). This overlap is not inherently problematic, as it reflects natural language settings, where syntactic and semantic structures are rarely novel and learning dynamics are influenced by previously learned distributions.

### 3.2.1 EVALUATION DATASETS

We use three datasets for factuality evaluation as summarized in Table 1 and described below. An exact list of topics included in these datasets is given in Appendix D.

**Wikipedia Science.** We collect 231 Wikipedia articles on science topics that were created after the cutoff date for GPT-J 6B's pre-training data, where we use keyword searching of Wikipedia meta-data to identify science articles. We focus on science topics as they contain detailed technical language, which is also common in sensitive data settings (e.g., clinical notes). While Wikipedia articles on these topics did not exist before 2020, we do expect some of the concepts in these articles to exist in other pre-training data sources, which makes it feasible for a DP model to produce facts on these topics, even without memorizing individual data points.

**Wikipedia AI.** We collect 124 Wikipedia articles on AI topics, where we hand-curate products and models that did not exist before 2020, along with related articles we expect to mention them. Unlike the Wikipedia Science articles, GPT-J 6B cannot have any knowledge of most of these concepts without fine-tuning, as they also could not have existed in other pre-training data sources. However, by constructing our data to contain articles that mention overlapping topics, we ensure that it is feasible for a DP model to learn them. For example, if our dataset only contained *DeepSeek (chatbot)*, DP would preclude learning of information isolated to one data point. By including *DeepSeek (chatbot)*, *DeepSeek*, and *DeepSeek (disambiguation)*, a DP model can hypothetically learn information about DeepSeek, as it is mentioned in multiple data points.

**Wikipedia pre-training.** For RQ2, where we investigate effects of DP fine-tuning on knowledge acquired during pre-training, we randomly sample 250 Wikipedia articles from the GPT-J pretraining data. We make sure that these articles are not included in fine-tuning dataset.

### 3.2.2 FINE-TUNING DATASET

Fine-tuning with differential privacy generally requires large enough datasets and large batch sizes (e.g. $\Omega(10^3)$ or more) (McMahan et al., 2018; Ponomareva et al., 2023). However, since our curated evaluation sets are insufficient to meet these batch size specifications, we intersperse our collected articles with an additional 20,000 randomly sampled Wikipedia articles that likely occurred in the pre-training data. We ensure that these samples do not overlap with the Wikipedia pre-training dataset used for factuality evaluation.

The data is divided into sequences of 1024 tokens, which is the unit of privacy protection. We fine-tune the model to produce an article when prompted on the article title (e.g., a topic). For evaluations, we similarly prompt the model with article titles and evaluate the factual accuracy of the generated text.

### 3.3 EVALUATION OF FACTUAL ACCURACY IN OPEN-ENDED TEXT GENERATION

Hallucinations are factual inaccuracies, so a higher factual accuracy is indicative of a lower hallucination rate. We evaluate the factual accuracy in open-ended text generation settings where we prompt the model with a title and generate a full Wikipedia article. We note that this is closer to the concurrent uses of LLMs, as opposed to traditional factuality evaluations based on cloze-style or short-form response queries designed to probe models (Youssef et al., 2023; Petroni et al., 2019). We use both automated and human assessments of factuality, as described below.

**Automated Evaluation via FactScore.** Given a generated text $d_i$, FactScore (Min et al., 2023) operates in two distinct phases: (i) **atomic claim extraction**, where $d_i$ is decomposed into a set of minimal, verifiable claims $\mathcal{AF}_{d_i}$, and (ii) **claim verification**, where each claim $\alpha_j^{(d_i)} \in \mathcal{AF}_{d_i}$ is evaluated for factuality using a verifier $\mathcal{V}$ conditioned on both intrinsic language model judgments and evidence retrieved from an external knowledge source $\mathcal{K}$, as detailed in Algorithm 1, Appendix F. We use the original Wikipedia article as the external knowledge source for verifying generated claims. As claim decomposition methods can produce redundant claims that artificially inflate scores, we use the CORE module (Jiang et al., 2024b) to filter down the superfluous and repetitive claims.

We use Llama-3.1-8B-Instruct for steps (i) and (ii) above involved in FactScore computations. We verify in Appendix I that the overall trends of FactScore results are consistent with other choices of models.

**Human Evaluation.** As FactScore is an automated metric relying on LLM judgments that may not be accurate, we conduct human evaluations to validate results and provide finer-grained analysis of outputted information. We use the Wikipedia AI dataset, as this dataset most carefully separates the pre-training and fine-tuning data. For the annotation task, we recruited 5 computer science graduate students, whom we expect to have high AI literacy. We ensure that the human evaluators see generations with both high and low factual accuracy using stratified sampling: we sample 15 articles with both DP ($\varepsilon = 16$) and non-DP models have FactScore $> 50\%$ and another 15 articles where both models have FactScore $< 50\%$. Our custom annotation interface (Appendix §E) displays a source text and a generated text side-by-side, followed by automatically decomposed atomic claims. Annotators rated claims for their overall veracity (correct/incorrect/unclear) and their support within the source text. Annotators were also asked to flag quality issues in decomposed claims, marking them as vague or subjective. Two annotators rated each claim, and we report the annotator agreement for both the non-DP and DP texts in Table 10 as Cohen's Kappa values for both the veracity of the claim, and whether or not it is supported by evidence in the training text corresponding to the topic it was generated for. [3]

### 3.4 ANALYSIS OF RECURRING HALLUCINATIONS

The metrics in §3.3 can capture if models hallucinate incorrect facts, but they do not distinguish between models that output a range of incorrect information (suggesting general noisiness) or if they repeatedly output the same incorrect information across generations (suggesting encoding of inaccurate facts). To identify recurring factual claims across model generations, we propose a multi-stage clustering algorithm. We give a high-level summary below, with precise pseudo-code in Algorithm 2, Appendix F.

Consider the generated synthetic documents grouped by the the topic $t \in \mathcal{T}$ used to prompt their generation. Each document is decomposed into atomic claims and these claims are aggregated into a claim set $C_t = c_i$ for each topic $t$.

We first index all extracted claims to their source (synthetic) documents, then cluster them using sentence-embedding–based agglomerative clustering to group semantically similar claims. To refine boundaries, we apply DBSCAN with Jaccard similarity, ensuring clusters are both semantically coherent and lexically consistent. This reduces cases where semantically related but factually distinct claims are grouped together.

Finally, we retain only recurring claim clusters, defined as clusters containing claims from at least two distinct documents for the same topic. This step isolates claims that recur across different generations, highlighting factual patterns the model consistently produces rather than one-off statements. We then analyze these clusters of claims to identify potential recurrent hallucinations.

---

[3] Annotators were offered compensation at an estimated payment rate of $20/hr

| Dataset | DP Setting | Average FS | Median FS | Q1 FS | Q3 FS | Avg Max FS / Topic | Avg Min FS / Topic | # of FS $\geq$0.5 | # of FS <0.5 |
|---|---|---|---|---|---|---|---|---|---|
| Wikipedia AI | $\varepsilon = \infty$ | 37.7 | 33.3 | 12.5 | 58.7 | 51.4 | 24.8 | 132 | 346 |
| | $\varepsilon = 16$ | 35.0 | 30.8 | 12.3 | 54.5 | 49.9 | 21.7 | 100 | 388 |
| | $\varepsilon = 8$ | **32.2** | **25.0** | 10.0 | **50.0** | **45.9** | **19.0** | 80 | **404** |
| | Base | 32.9 | 27.3 | **9.1** | **50.0** | 47.5 | 19.3 | **78** | 362 |
| Wikipedia Science | $\varepsilon = \infty$ | 56.1 | 58.3 | 33.3 | 78.9 | 70.3 | 41.6 | 393 | 495 |
| | $\varepsilon = 16$ | 54.9 | 57.1 | 30.0 | 80.0 | 70.1 | 39.3 | 323 | 555 |
| | $\varepsilon = 8$ | 53.2 | 55.6 | 28.6 | 77.8 | **68.5** | 37.9 | 290 | **592** |
| | Base | **52.1** | **50.0** | **27.3** | **75.0** | 76.0 | **27.5** | **190** | 435 |

Table 2: FactScore (FS; reported as a percentage) results from GPT-J 6B, fine-tuned with different DP budgets, evaluated at temperature $\tau = 0.3$. Reported are average, median, and quartile FactScores, per-topic average maxima and minima, and counts of factual ($\geq 0.5$) and non-factual ($< 0.5$) responses. "Base" refers to the base GPT-J 6B model with no fine-tuning. The row with the lowest FactScore (i.e. most hallucinations) is bolded. The factual consistency decreases with stricter privacy budgets (smaller $\varepsilon$), highlighting the tradeoff between privacy and hallucinations.

## 4 EXPERIMENTAL RESULTS

We empirically study each of the three research questions in turn.

### 4.1 RQ1: HALLUCINATION CONCERNING FINE-TUNING FACTS IN DP FINE-TUNED MODELS

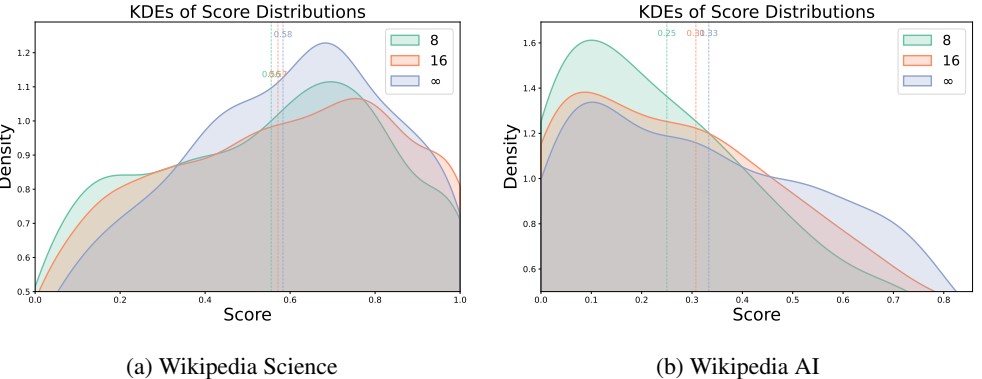

(a) Wikipedia Science       (b) Wikipedia AI

Figure 1: Kernel density (KDE) plots comparing FactScore distributions when finetuning GPT-J 6B under different DP budgets. As the privacy budget becomes more stringent ($\varepsilon$ from $\infty$ to 8), the FactScore distribution shifts leftward, indicating that stronger privacy constraints result in less factually consistent outcomes.

**Automated Evaluation.** The automated factuality evaluation for unseen fine-tuning data in Table 2 suggests that DP fine-tuning leads to more frequent hallucinations. Specifically, we observe the following general trend in factuality: non-DP ($\varepsilon = \infty$) $\succ$ DP ($\varepsilon = 16$) $\succ$ DP ($\varepsilon = 8$) $\succ$ base ($\varepsilon = 0$), where "base" denotes the pre-trained model. That is, we observe a consistent decrease in the average FactScore for the text generated from models fine-tuned with DP compared to those trained without DP. This decline is more pronounced for the stricter privacy budget setting $\varepsilon = 8$, and is further illustrated by the skew (toward lower factuality scores) in the distributions of FactScores for DP-trained models in Figure 1. Furthermore, the proportion of factual summaries (FactScore $\geq 0.5$) also progressively decreases with stricter privacy budgets.

As a sanity check, observe that the model trained without DP outputs a higher percentage of correct claims than the base model (32.9% → 37.7% for the AI articles and 52.1% → 56.1% for the Science articles), demonstrating the usefulness of fine-tuning. Notably, on the Wikipedia AI dataset, the model fine-tuned with $\varepsilon = 8$ performs worse or comparably (mean: 32.2%; median: 25.0%), to the base model (mean: 32.9%; median: 27.3%), despite the training loss decreasing progressively when fine-tuning with this privacy budget. While this marginal degradation in FactScore

| Wikipedia AI | FactScore < 0.5 | | FactScore ≥ 0.5 | |
|---|---|---|---|---|
| | $\varepsilon = \infty$ | $\varepsilon = 16$ | $\varepsilon = \infty$ | $\varepsilon = 16$ |
| Avg. Veracity Score | 0.18 | 0.17 | 0.51 | 0.43 |
| Avg. Support Ratio | 0.19 | 0.09 | 0.54 | 0.32 |
| Quality Issues (count) | 87 | 79 | 55.5 | 79 |

Table 3: Comparison of human annotation results on Wiki AI FactScore summaries at $\varepsilon = \infty$ and $\varepsilon = 16$, stratified by response groups with veracity scores below 0.5 and above 0.5. Reported metrics are average veracity score, average support ratio, and number of claims annotated as having quality issues. In both cases, we observe an increase in hallucinations with DP ($\varepsilon = 16$) relative to no DP ($\varepsilon = \infty$).

| | | Supported | | | Unsupported | | | |
|---|---|---|---|---|---|---|---|---|
| | DP Setting | Recur Count | Total Count | % Avg Recur | Recur Count | Total Count | % Avg Recur | Supported: Unsupported |
| Wikipedia AI | $\varepsilon = \infty$ | 242 | 1198 | 20.20 | 494 | 1992 | 24.80 | 0.490 |
| | $\varepsilon = 16$ | 200 | 1103 | 18.13 | **523** | 2002 | **26.12** | 0.382 |
| | $\varepsilon = 8$ | **174** | 1019 | **17.08** | 509 | 2011 | 25.31 | **0.342** |
| Wikipedia Science | $\varepsilon = \infty$ | 705 | 2896 | 24.34 | 560 | 2417 | 23.17 | 1.259 |
| | $\varepsilon = 16$ | 582 | 2467 | 23.59 | 556 | 2128 | 26.13 | 1.047 |
| | $\varepsilon = 8$ | **556** | 2378 | **23.38** | 633 | 2222 | **28.49** | 0.878 |

Table 4: Analysis of recurrent claims and hallucinations (§3.4) for temperature $\tau = 0.3$. "Total Count" reports the total number of generated clusters, "Recur Count" reports the number of those clusters with $\geq 2$ supporting documents, and %Avg Recur is "Recur Count"/"Total Count". The far right column reports "Recur Count" of supported claims / "Recur Count" of unsupported claims. DP models output a lower ratio of repeated supported claims to unsupported claims, suggesting increased repeated hallucinations. Bolding indicates most hallucinations (i.e. least supported or most unsupported).

for Wikipedia AI articles could be statistical noise, it could also indicate the gradient signal from the AI articles being masked by the larger levels of noise introduced by DP-SGD when fine-tuning with lower privacy budgets ($\varepsilon = 8$). Under a more generous budget ($\varepsilon = 16$), FactScore does increase compared to the baseline (mean: 35.0%), but still falls short of the model trained without DP (mean: 37.7%).

FactScore is higher for the Science articles compared to AI articles; median scores are consistently $\geq 50\%$. Furthermore, the DP model with $\varepsilon = 8$ achieves a higher mean FactScore than the base model, suggesting that DP fine-tuning hallucinates less than the base model. We again observe a clear privacy-hallucination tradeoff where FactScore decreases with more stringent privacy budgets (smaller $\varepsilon$). The differences between the AI and Science datasets likely results from their differing overlap with pre-training data: concepts from Science articles are more likely to occur in pre-training data, even if exact articles are non-overlapping.

**Human Evaluation.** Table 3 reports the average ratings selected by human annotators. In lower-quality generations (FactScore $\leq 0.5$), annotators rated both the non-DP and DP models with equally low veracity, while the DP model exhibited more unsupported facts. In higher-quality generations (FactScore $> 0.5$), annotators rated the DP model as outputting both lower veracity information and more unsupported facts. There are not conclusive differences in counts of quality issues. While agreement between human annotators was generally high, model-human agreement was not always high (Table 10), indicating limitations of relying exclusively on automated evaluation. Regardless, overall trends are consistent between human and automated evaluations: both indicate greater hallucination in the DP model, even the model with a more generous privacy budget.

**Recurring Hallucination Analysis.** Table 4 reports results from the claim clustering analysis, as described in §3.4. In both datasets, the DP model with $\varepsilon = 8$ outputs fewer recurring supported claims (i.e. factually correct statements) than other models. DP models also output more recurring unsupported claims than non-DP models, with $\varepsilon = 8$ highest for Wikipedia Science and $\varepsilon = 16$ highest for Wikipedia AI. For both datasets, the ratio of supported recurring claims to unsupported recurring claims is consistently lower with stricter privacy budgets. This indicates that the hallucination-

| $\varepsilon = \infty$ | 'AlphaEvolve is used for generating the molecular structures of organic molecules.', 'AlphaEvolve is used for generating molecular structures.' |
|---|---|
| $\varepsilon = 16$ | 'There are two types of enemies in the game.', 'The game features two types of enemies.' |
| $\varepsilon = 8$ | 'Black Hole Interactive is a game development company.', 'Black Hole Interactive is a video game development company.' |

Table 5: Example unsupported claim clusters (hallucinations) for each model. We provide additional examples in Appendix P.

| DP Budget | Average FS | Median FS | Q1 FS | Q3 FS | Avg Max FS / Topic | Avg Min FS / Topic | # of FS $\geq$ 0.5 | # of FS < 0.5 |
|---|---|---|---|---|---|---|---|---|
| $\varepsilon = \infty$ | 30.4 | 23.5 | 9.1 | 46.2 | 42.5 | 18.8 | 131 | **841** |
| $\varepsilon = 16$ | **29.5** | **23.1** | 9.1 | **44.4** | 41.8 | **18.1** | **125** | 829 |
| $\varepsilon = 8$ | 30.6 | 25.0 | **8.3** | 50.0 | 43.2 | 18.8 | 144 | 824 |
| Base | 29.7 | 25.0 | 9.1 | **44.4** | **41.0** | 19.6 | 139 | 837 |

Table 6: FactScores (FS; reported in %) for GPT-J evaluated with temperature $\tau = 0.3$ over Wikipedia pre-training, which contains articles likely to be in pre-training data, but not included in fine-tuning. DP models perform similarly as non-DP models, suggesting no disruption to facts learned in pre-training. Bolding indicates worse FactScores (i.e. higher hallucinations).

privacy tradeoff involves groups of similar factual inaccuracies, suggesting a systemic encoding of inaccurate facts, rather than random hallucinations.

In Table 5, we show an example of a cluster of recurring hallucinations for each model. Here, the generation prompt was "AlphaEvolve." The non-DP model correctly outputs that AlphaEvolve is a model, but incorrectly describes model use. In contrast, the DP models both hallucinate that AlphaEvolve is a video game, with repeated fabricated information about the development and game play.

### 4.2 RQ2: Hallucination of pre-training facts in DP fine-tuned models

Next, we turn to RQ2 to address whether DP fine-tuning degrades the knowledge already encoded during standard pre-training. Table 6 reports FactScores over Wikipedia pre-training. The differences between DP and non-DP models are marginal, suggesting DP finetuning does not disrupt factual knowledge acquired from pre-training data. This trend is consistent across temperatures (Figure 6; Appendix) and stands in stark contrast to the evaluations on previously unseen data, where stronger privacy constraints correlate with lower factual accuracy.

### 4.3 RQ3: Hallucinations in DP pre-trained models

Finally, we investigate if DP pre-training, as opposed to fine-tuning, leads to increased hallucinations in Table 7. It is worth noting that the pre-training data mixtures for the Gemma models likely include *all* the datasets used in our factual evaluations, while we expect GPT-2 to only have been trained on the Wikipedia pre-training evaluation set.

The DP pre-trained model (VaultGemma) consistently outputs a greater percentage of inaccurate facts relative to the non-DP pre-trained Gemma3-1B. On the Wikipedia pre-training data, Gemma3 achieves an average FactScore of 26.6, compared to 22.0 for VaultGemma. This difference is reflected further in the proportion of factual summaries, with Gemma3-1B producing over twice as many factually correct texts as VaultGemma. This gap in factual correctness is more pronounced for the domain-specific scientific and AI articles, where the difference Gemma3-1B and Vault-Gemma reaches 9.9 points (50.4 vs. 39.5) and 8.7 points (51.9 vs. 43.2), respectively. Distributions of FactScores (Figure 2) show the same results: VaultGemma outputs have a higher density of low FactScores than Gemma3-1B.

Compared to GPT-2 XL, VaultGemma does have higher average FactScores for AI and pre-training articles. As the cutoff date for GPT-2 XL's pre-training data was in 2019, this model had no exposure to most concepts in the AI articles, thus constituting an extremely low bar for factual correctness in this setting. The improved FactScores of VaultGemma over the Wikipedia pre-training may be a reflection of general improvements in LLM development over the last 5 years that are not undone by DP training. More surprisingly, VaultGemma fails to output more factually correct information than GPT-2 XL in the Wikipedia Science setting, even though these articles were all created after 2020, suggesting they were included in VaultGemma training data and not GPT-2 XL data.

| Dataset | Pretrained Model | Average FS | Median FS | Q1 FS | Q3 FS | Avg Max FS / Topic | Avg Min FS / Topic | # of FS ≥ 0.5 | # of FS <0.5 |
|---|---|---|---|---|---|---|---|---|---|
| AI | Gemma3-1B-PT | 51.9 | 55.6 | 25.0 | 79.6 | 65.8 | 37.2 | 222 | 248 |
| | VaultGemma-1B | **43.2** | 43.3 | 17.0 | 66.7 | 59.9 | 27.7 | 126 | 340 |
| | GPT-2-1.5B | 27.7 | **22.2** | **7.6** | **45.5** | **45.2** | **13.5** | **38** | **366** |
| Science | Gemma3-1B-PT | 50.4 | 50.0 | 25.0 | 75.0 | 72.7 | 28.8 | 219 | 600 |
| | VaultGemma-1B | 39.5 | **33.3** | **14.6** | 60.0 | 65.2 | **14.8** | **81** | 697 |
| | GPT-2-1.5B | **39.1** | **33.3** | 16.9 | **57.1** | 61.3 | 17.9 | 84 | **750** |
| Pretraining | Gemma3-1B-PT | 26.6 | 17.6 | 7.1 | 42.9 | 39.8 | 14.7 | 140 | 835 |
| | VaultGemma-1B | 22.0 | 14.3 | **0.0** | 33.3 | 37.2 | 9.5 | 58 | **931** |
| | GPT-2-1.5B | **17.6** | **11.1** | **0.0** | **25.0** | **31.6** | **6.0** | **32** | 880 |

Table 7: FactScore (FS; reported as %) results of pre-trained models, evaluated at temperature $\tau = 0.3$. Reported are average, median, and quartile FactScores, per-topic average maxima and minima, and counts of factual ($\geq 0.5$) and non-factual ($<0.5$) responses. Bolding indicates lower FactScore (increased hallucinations).

Overall, these results suggest DP-SGD pre-training can significantly weaken a model's ability to encode and output factually correct information, with DP model outputs sometimes as hallucinated as outputs from a model never directly exposed to the targeted information.

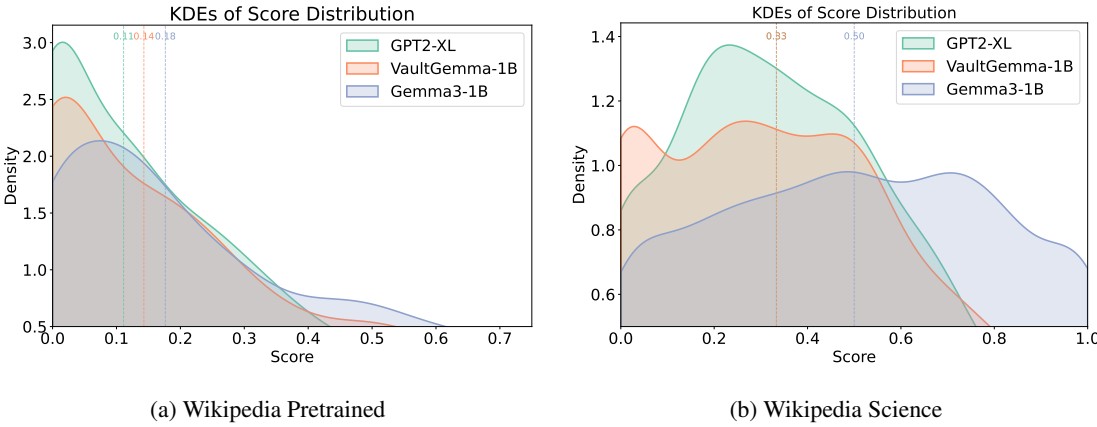

(a) Wikipedia Pretrained                (b) Wikipedia Science

Figure 2: Kernel density (KDE) plots comparing FactScore distributions for models pre-trained with (VaultGemma $\varepsilon = 2$) and without DP (Gemma3-1B & GPT2-XL). GPT2-XL, which has not seen the data during training, and VaultGemma (DP pre-trained) both lag behind Gemma-3B in factual consistency.

## 5 DISCUSSION AND CONCLUSIONS

We reveal and systematically analyze a privacy–hallucination tradeoff in differentially private language models. Our results show that differential privacy significantly hinders the acquisition of new factual associations during pre-training and fine-tuning, leading to hallucinations. Further, we find the same unsupported facts repeated across generations, suggesting a systematic encoding of inaccurate facts rather than random errors. While our metrics focus on analyzing model outputs, they suggest the clipping and additive noise in DP may lead to the acquisition of incorrect facts; this could potentially be targeted directly in future work.

Our findings suggest that, in some applications, DP fine-tuning may simply be unsuitable. For example, a healthcare model trained without DP on public data, like medical journal articles, may be more reliable than one trained with DP on patient data, even if the non-DP model is unable to assist with some tasks or queries. Use of DP in these settings requires careful analysis of if the perceived benefits (e.g., acquisition of domain-specific knowledge) are worth the risk of increased hallucinations.

Alternatively, in settings where training on private data is essential, more research is needed to counteract hallucinations. While fully mitigating hallucinations is an unsolved problem even in non-DP settings, targeted interventions could reduce risks. Specialized post-training objectives could steer models towards expressing uncertainty or refusal over generating incorrect content (Kalai et al., 2025). A nascent line of work has proposed incorporating sensitive information through DP retrieval-augmented generation, rather than fine-tuning (Koga et al., 2024; Grislain, 2025), though maintaining a privacy guarantees under multiple queries remains challenging, and more research is needed to assess if hallucinations persist in these settings.

Overall, our findings also underscore that existing DP training methods can compromise factual reliability in high-stakes applications. This highlights the urgent need for refined privacy-preserving approaches that balance rigorous privacy guarantees with factual accuracy. Addressing this tradeoff is essential for deploying trustworthy AI systems in sensitive domains such as healthcare and law where both privacy and factual consistency are non-negotiable.

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

## A    BACKGROUND: DIFFERENTIAL PRIVACY

Differential privacy offers a formal privacy guarantee that ensures that any individual's data cannot be inferred from a query applied to a dataset (Dwork et al., 2006). In other words, the result of such a query is nearly indistinguishable from the result of the same query applied to a dataset that either includes a modified version of the individual's data or excludes the record entirely, thereby preserving the individual's privacy. In this case, the notion of adjacency specifies exactly how changing single record in the original dataset $D$ yields the modified dataset $D'$.

We review the definition of DP here; we refer to the textbooks (Dwork et al., 2014; Fioretto et al., 2025) for specific details and the guide (Ponomareva et al., 2023) for details on DP-SGD.

Formally, differential privacy is defined as follows:

**Definition**: Two datasets $D$ and $D'$ are said to be neighboring (in the add-or-remove sense) if $D = D' \cup \{x\}$ or $D' = D \cup \{x\}$ for some training example $x$. A randomized algorithm $A$ is $(\varepsilon, \delta)$-private for some $\varepsilon > 0$ and $\delta \in [0, 1]$ if for *any* two neighboring datasets $D, D'$, the following holds true for all sets $Y$ in the range of $A$:
$$\Pr[A(D) \in Y] \le e^{\varepsilon} \Pr[A(D') \in Y] + \delta.$$
The value of $\varepsilon$ denotes the privacy budget, while $\delta$ specifies the likelihood that the privacy guarantee may fail. If $\delta$ is set to 0, this implies a purely differentially private setting with no probability of the guarantee being broken. The value of $\varepsilon$ constrains how similar the outputs of both distributions are; a higher $\varepsilon$ value indicates a greater privacy budget, meaning the algorithm is less private. DP guarantees that even if an adversary has access to any side-knowledge, the privacy leakage of $(\varepsilon, \delta)$-DP algorithms will not increase. Additionally, another property of DP is that it ensures that any post-processing on the outputs of $(\varepsilon, \delta)$-differentially private algorithms will remain $(\varepsilon, \delta)$-differentially private.

We use DP-SGD (Abadi et al., 2016), a modification to the stochastic gradient descent (SGD) algorithm, which is typically used to train neural networks. DP-SGD clips the gradients to limit the contribution of individual samples from the training data and subsequently adds noise from the Gaussian distribution to the sum of the clipped gradients across all samples.[4] DP-SGD thus provides a differentially private guarantee to obfuscate the gradient update, thereby ensuring that the contribution of any given sample in the training data is indistinguishable due to the aforementioned post-processing property. This process ensures $(\varepsilon, \delta)$-differential privacy for each model update. Given a privacy budget, number of epochs, and other training parameters, we can estimate the privacy parameters using standard privacy accounting algorithms, which implemented in common software.

## B    EXPERIMENTAL SETUP

We conduct both standard (non-private) fine-tuning and differentially private (DP) fine-tuning, and the noise multiplier in the DP setting is calculated using the PLD accountant. We report the results for two privacy budgets: $\varepsilon = 8$ and $\varepsilon = 16$, and the $\delta$ is set to $1/n^{1.1}$, where $n$ is the size of the training set. We use the PLD accounting algorithm proposed in Doroshenko et al. (2022), which provides us with tighter estimates of the privacy loss as compared to alternate accounting techniques. This in turn allows us to more accurately determine the noise multiplier required to satisfy the specified privacy budget for our fine-tuning setups.

The hyperparameters are summarized as below.

**LoRA Gradient Update**    We apply Low-Rank Adaptation (LoRA) (Hu et al., 2022) in all experiments. The adapted weight is parameterized as
$$W = W_0 + \frac{\alpha}{r} AB$$
where the LoRA matrices are $A \in \mathbb{R}^{d \times r}$, $B \in \mathbb{R}^{r \times k}$, and $W_0$ is the frozen pre-trained weight. During training, the LoRA parameters $A, B$ are updated via:
$$(A^{(t+1)}, B^{(t+1)}) = (A^{(t)}, B^{(t)}) - \eta \cdot \mathrm{clip}\left(\nabla_{A,B} L, c\right)$$

---

[4]The noise added is independent coordinate-wise and across time, although DP-SGD with temporal correlations has recently been of growing interest (Pillutla et al., 2025).

| Setting | Batch Size | Epochs | LR | Clip | LoRA $r$ | LoRA $\alpha$ | Seq. Len. | DP/PLD |
|---|---|---|---|---|---|---|---|---|
| Vanilla | 8 | 15 | $1 \times 10^{-4}$ | 1.0 | 4 | 32 | 512 | – |
| DP, $\varepsilon \in \{8, 16\}$ | 4096 | 20 | $1 \times 10^{-4}$ | 1.0 | 4 | 512 | 512 | $\delta = 1/n^{1.1}$ |

Table 8: Summary of major hyperparameters in vanilla and private training settings.

where $\eta$ is the learning rate, $c = 1.0$ is the clipping norm, and $L$ is the loss function. In the DP setting, noise is added to the clipped gradient.

During training, the update step for $(A, B)$ in SGD or DP-SGD is:

$$(A^{(t+1)}, B^{(t+1)}) = (A^{(t)}, B^{(t)}) - \eta \cdot \widetilde{\nabla} L$$

where

$$\widetilde{\nabla} L = \text{clip}(\nabla L, c) + \mathcal{N}(0, \sigma^2)$$

$$\nabla_A L = \frac{\alpha}{r} \nabla_W L \cdot B^\top$$

Instead of using a large learning rate in the DP fine-tuning, we use a larger value of ($\alpha = 512$) in the DP setting (compared to ($\alpha = 32$) in the standard fine-tuning setting) to amplify the contribution of the adapted weights without increasing the noise magnitude (which is applied to the gradients directly). This prevents the gradient signal from being obscured by the DP noise.

## C  DATASET : EXAMPLES OF INPUT

Table 9: Examples of excerpts of the input and output pairs from our Wikipedia AI and Science articles.

| Input | Output |
|---|---|
| Right to Know | Right To Know is a non profit support project for those who discover via genealogical genetic testing that their lineage is not what they had supposed it to be due to family secrets and misattributed parentage, thus raising existential issues of adoption, race, ethnicity, culture, rape, etc.  == See also == Genealogy Genetic testing == External links == Right To Know - Your Genetic Identity. |
| Neurosemiotics | Neurosemiotics is an area of science which studies the neural aspects of meaning making. It interconnects neurobiology, biosemiotics and cognitive semiotics. Neurolinguistics, neuropsychology and neurosemantics can be seen as parts of neurosemiotics. == Description == The pioneers of neurosemiotics include Jakob von Uexküll, Kurt Goldstein, Friedrich Rothschild, and others. The first graduate courses on neurosemiotics were taught in some American and Canadian universities since 1970s. The term 'neurosemiotics' is also not much older. Neurosemiotics demonstrates which are the necessary conditions and processes responsible for semiosis in the neural tissue. It also describes the differences in the complexity of meaning making in animals of different complexity of the nervous system and the brain.  == See also == Semiotics Zoosemiotics. |

| Input | Output |
| --- | --- |
| Cyclosiloxanes | Cyclosiloxanes are a class of silicone material. They are volatile and often used as a solvent. The three main commercial varies are octamethylcyclotetrasiloxane (D4), decamethylcyclopentasiloxane (D5) and dodecamethylcyclohexasiloxane (D6). They evaporate and degrade in air under sunlight. == Octamethylcyclotetrasiloxane (D4) == The octamethylcyclotetrasiloxane silicone liquid has no odor and consists of four repeating units of silicon (Si) and oxygen (O) atoms in a closed loop giving it a circular structure. Each silicon atom has two methyl groups attached (CH3). == Decamethylcyclopentasiloxane (D5) == Decamethylcyclopentasiloxane silicone liquid has no odor and consists of five repeating units of silicon (Si) and oxygen (O) atoms in a closed loop giving it a circular structure. Each silicon atom has two methyl groups attached (CH3). Typically it is used as an ingredient in antiperspirant, skin cream, sun protection lotion and make-up. With a low surface tension of 18 mN/m this material has good spreading properties. |
| Cancer exodus hypothesis | The cancer exodus hypothesis establishes that circulating tumor cell clusters (CTC clusters) maintain their multicellular structure throughout the metastatic process. It was previously thought that these clusters must dissociate into single cells during metastasis. According to the hypothesis, CTC clusters intravasate (enter the bloodstream), travel through circulation as a cohesive unit, and extravasate (exit the bloodstream) at distant sites without disaggregating, significantly enhancing their metastatic potential. This concept is considered a key advancement in understanding of cancer biology and CTCs role in cancer metastasis. == Mechanism == Traditionally, it was believed that CTC clusters needed to dissociate into individual cells during their journey through the bloodstream to seed secondary tumors. However, recent studies show that CTC clusters can travel through the bloodstream intact, enabling them to perform every step of metastasis while maintaining their group/cluster structure. |
| Generative pre-trained transformer | Generative Pre-trained Transformer 1 (GPT-1) was the first of OpenAI's large language models following Google's invention of the transformer architecture in 2017. In June 2018, OpenAI released a paper entitled "Improving Language Understanding by Generative Pre-Training", in which they introduced that initial model along with the general concept of a generative pre-trained transformer. Up to that point, the best-performing neural NLP models primarily employed supervised learning from large amounts of manually labeled data. This reliance on supervised learning limited their use of datasets that were not well-annotated, in addition to making it prohibitively expensive and time-consuming to train extremely large models; many languages (such as Swahili or Haitian Creole) are difficult to translate and interpret using such models due to a lack of available text for corpus-building. In contrast, a GPT's "semi-supervised" approach involved two stages: an unsupervised generative "pre-training" stage in which a language modeling objective was used to set initial parameters, and a supervised discriminative "fine-tuning" stage in which these parameters were adapted to a target task. |
| GPTZero | GPTZero is an artificial intelligence detection software developed to identify artificially generated text, such as those produced by large language models. While GPTZero was praised for its efforts to prevent academic dishonesty, many news outlets criticized the tool's false positive rate, which can be especially harmful in academic settings. == History == GPTZero was developed by Edward Tian, a Princeton University undergraduate student, and launched online in January 2023 in response to concerns about AI-generated usage in academic plagiarism. GPTZero said in May 2023 it raised over 3.5 million dollars in seed funding. In the first week of its release, the GPTZero experienced 30,000 uses, which led to a crash. It was supported by the web app company Streamlit, who allocated more server resources in response. In July 2024, it had 4 million users, compared to 1 million one year earlier. In summer 2024, GPTZero raised $10 million in Series A round funding. In September 2024, GPTZero announced an authorship tracking software that enables "to compile and share data about their writing process such as their copy/paste history, the number of editors they had, and how long editing took", in an effort "to move away from an all-or-nothing paradigm around AI writing towards a more nuanced one." |

| Input | Output |
|---|---|
| GPT 4.5 | GPT-4.5 (codenamed "Orion") is a large language model developed by OpenAI as part of the GPT series. Officially released on February 27, 2025, GPT-4.5 is available to users subscribed to the ChatGPT Plus and Pro plans across web, mobile, and desktop platforms. Access is also provided through the OpenAI API and the OpenAI Developer Playground. == Overview == It was primarily trained using unsupervised learning, which improves its ability to recognize patterns, draw connections, and generate creative insights without reasoning. This method was combined with supervised fine-tuning and reinforcement learning from human feedback. The computational resources needed for training were provided by Microsoft Azure. Sam Altman described GPT-4.5 as a "giant, expensive model". |
| Claude | Claude is a family of large language models developed by Anthropic. The first model was released in March 2023. The Claude 3 family, released in March 2024, consists of three models: Haiku, optimized for speed; Sonnet, which balances capability and performance; and Opus, designed for complex reasoning tasks. These models can process both text and images, with Claude 3 Opus demonstrating enhanced capabilities in areas like mathematics, programming, and logical reasoning compared to previous versions. Claude 4, which includes Opus and Sonnet, was released in May 2025. == Training == Claude models are generative pre-trained transformers. They have been pre-trained to predict the next word in large amounts of text. Then, they have been fine-tuned, notably using constitutional AI and reinforcement learning from human feedback (RLHF). |

## D  LIST OF TOPICS IN THE DATASETS

### D.1  WIKIPEDIA AI

**124 articles**: DALL-E; OpenAI; Midjourney; Imagen (text-to-image model); Text-to-image model; Recraft; DeepSeek; DeepSeek (chatbot); Liang Wenfeng; High-Flyer; 2025 in artificial intelligence; DeepSeek (disambiguation); Six Little Dragons; R1; Ideogram (text-to-image model); Stable Diffusion; Automatic1111; ComfyUI; Stability AI; Emad Mostaque; Artificial intelligence and copyright; Fooocus; LAION; Sai; BLOOM (language model); Gemini; Gemini (chatbot); Gemini (language model); Gemini Robotics; Gemini Home Entertainment; Jet Force Gemini; Pixel 9; NotebookLM; Large language model; AlphaEvolve; Anthropic; Google Lens; Google ai stodio; Android XR; Chris Welty; Large language models in government; ChatGPT; Generative pre-trained transformer; GPT-4; GPT; GPT-4o; GPT-3; GPT-2; GPT-4.1; GPT-4.5; GPT-1; AutoGPT; Microsoft Copilot; GPTs; GPT-J; GPT Store; OpenAI o1; GPT4-Chan; GPTZero; Sora (text-to-video model); EleutherAI; YandexGPT; Writesonic; ChatGPT in education; Pause Giant AI Experiments: An Open Letter; Deep Learning (South Park); PauseAI; Chinchilla (language model); Artificial intelligence content detection; General-purpose technology; The Last Screenwriter; Wu Dao; Microsoft Recall; Alice and Sparkle; Amazon Q; Connor Leahy; Multimodal learning; OpenAI o4-mini; 2022 in artificial intelligence; Death of an Author (novella); GigaChat; P(doom); XLNet; Boyfriend Maker; 2023 in artificial intelligence; LLMs in higher education; Perceiver; NovelAI; Supremacy (book); Rabbit r1; Preamble (company); BookCorpus; Omneky; Machine unlearning; Artificial empathy; Llama (language model); Llama.cpp; DBRX; Llama (disambiguation); Alpaca (disambiguation); Qwen; Brave Leo; B65; Mistral; Mistral AI; Arthur Mensch; General Catalyst; Cédric O; Le Chat (disambiguation); PaLM; List of large language models; Prompt engineering; Foundation model; BERT (language model); LaMDA; T5 (language model); Alibaba Group; Claude (language model); Grok (chatbot); XAI (company); Colossus (supercomputer); X Corp.; Explainable artificial intelligence; Google DeepMind

### D.2  WIKIPEDIA SCIENCE

**231 articles**: Eurotrac; Scienticide; 505(b)(2) regulatory pathway; Anti-asthmatic agent; Breastmilk medicine; Cancer exodus hypothesis; Confocal endoscopy; Diabetes self-management; Dorsal pancreatic agenesis; Drone-Enhanced Emergency Medical Services; Electronic health record (Germany); Follicular drug delivery; LAMA2 related congenital muscular dystrophy; Most Favored Nation Drug Pricing; Musicians' Medicine; Poison exon; RNU2-2 syndrome; RNU4-2 syndrome; Synthetic Cannabinoid Use Disorder; Urinary anti-infective agent; Vestibular paroxysmia; Antarlide; Bioliteracy; Cancer exodus hypothesis; Dermestarium; Functional information; Interdigitation; Plasmagene; Poison exon; Polylecty; Spatial biology; Edge states; Electrostatic solitary wave; Frenesy (physics); History of the LED; HUN-REN Wigner Research Centre for Physics; Joaquim da Costa Ribeiro; Missile lofting; Nottingham effect;

Physics of Life; Quasi-isodynamic stellarator; Riccardo D'Auria (theoretical physicist); Shockwave cosmology; Synchronous lateral excitation; Toroidal solenoid; Wohlfarth Lectureship; Compliance constants; Cononsolvency; Corrosion inhibitors for the petroleum industry; Cyclosiloxane; Dark oxygen; Direct reduction; Energy-rich species; Grupo Fertiberia; Intrinsic DNA fluorescence; Krupp–Renn process; Mental gland; Probico; School of Molecular Sciences; Shape of the atomic nucleus; Stable phosphorus radicals; Superelectrophilic anion; TOP Assay; Mathematical oncology; Mathethon; The Math(s) Fix; Conductivity cell; Generalized renewal process; Glossary of engineering: M–Z; Marine construction; Museum of Engines and Mechanisms; Northern Technical College; Safer end of engineering life; Synchronous lateral excitation; The Clark Collection of Mechanical Movements; Third medium contact method; UNESCO World Engineering Day for Sustainable Development; Positive health; Bell's mania; Chialvo map; Dysfunctome; Femoral nerve dysfunction; Fiber photometry; Fork cell; High Price (book); Hyper-empathy; Large dense core vesicles; Lateral olfactory tract usher substance; Malaria therapy; Max Planck Institute for Biological Intelligence; Nerve glide; Neural synchrony; Neurosemiotics; Neurotrophin mimetics; Optogenetic methods to record cellular activity; Personality neuroscience; Representational drift; Single-particle trajectory; Smell training; Spongy degeneration of the central nervous system; Walk Again Project; Amoeboflagellate; Borg (microbiology); Chrompodellid; Dissimilatory iron reducing bacteria; Garrod Lecture and Medal; Hydrocarbonoclastic bacteria; Laboratory-acquired infection; Matground; Microbial pathogenesis; Milnesium alpigenum; Mitochondrion-related organelle; Phageome; Phytoplankton microbiome; Virivore; Virome analysis; Zodletone Mountain; Glossary of cellular and molecular biology (M–Z); Agricultural weed syndrome; Cell autonomous sex identity; Codon reassignment; De novo domestication; Endemixit; Genetic map function; Hovlinc; Integrative and conjugative element; Jena Declaration; Macrosatellite; Museomics; Poison exon; Polydactyly-myopia syndrome; Red cell genotyping; Right To Know; Selection limits; Shadow effect; Transcriptome-wide association study; Tumor mutational burden; Allogeneic processed thymus tissue; Cellular anastasis; COVID-19 passports in the United Kingdom; History of phagocytosis; Immunocapitalism; Macrophage-activating lipopeptide 2; Metal allergy; Milk immunity; Myocarditis-myositis-myasthenia gravis overlap syndrome; Oligoclonal antibody; P-i mechanism; Pathogen avoidance; Peripheral ulcerative keratitis; Post-acute infection syndrome; RVT-802; T memory stem cell; Thymic mimetic cells; TMEM61; Type 2 inflammation; Vaccine passports during the COVID-19 pandemic; Vaccine resistance; Zigakibart; 2022–2023 pediatric care crisis; Acoustic epidemiology; Causal pie model; Connecting Organizations for Regional Disease Surveillance; Elimination of tuberculosis; Epidemics Act; Epidemiology in Relation to Air Travel; Epidemiology of gonorrhoea; European Society of Health and Medical Sociology; Harvard Six Cities study; Hyperendemic; Loneliness epidemic; Microbial pathogenesis; Origin tracing; Pathogenic microorganisms in frozen environments; SARS-CoV-2 in white-tailed deer; Source attribution; Sporadic disease; Outline of public health; Alcohol tax; Autobesity; Biomedical Research Center; CalOptima; Care Group approach; Christian Health Association of Malawi; Commercial determinants of health; Connecting Organizations for Regional Disease Surveillance; COVID-19 lockdowns by country; Epidemics Act; History of public health in Australia; History of public health in Canada; History of public health in Chicago; History of public health in New York City; History of public health in the United Kingdom; History of public health in the United States; Intermittent water supply; International Association for Cannabinoid Medicines; Langya virus; LGBT life expectancy; User:Lguzmang06/sandbox; Loneliness epidemic; Malawi Network of AIDS Services; Mass. and Cass; Medical officer of environmental health; Motonormativity; National Association for People living with HIV/AIDS in Malawi; North Karelia Project; Nuisance ordinance; Origin tracing; Preventive and social medicine; Responsibility Deal; SaTScan; Sleeping Sickness Commission; Slug gate; Social determinants of mental health; Special Programme of Research, Development and Research Training in Human Reproduction- HRP; Telemedicine in Nepal; Vaccine equity; Vaccine line jumping; Vaccine storage; WHO Hub for Pandemic and Epidemic Intelligence; WHO public health prizes and awards; Additive effect; Antica Farmacia Sant'Anna; FK962; Institute for Safe Medication Practices; Model-Informed Precision Dosing; P-i mechanism; Penetration enhancer; Pharmacological cardiotoxicity; Pullulan bioconjugate; Reversible Hill equation

## E MANUAL ANNOTATIONS: CLAIM ANNOTATION INTERFACE

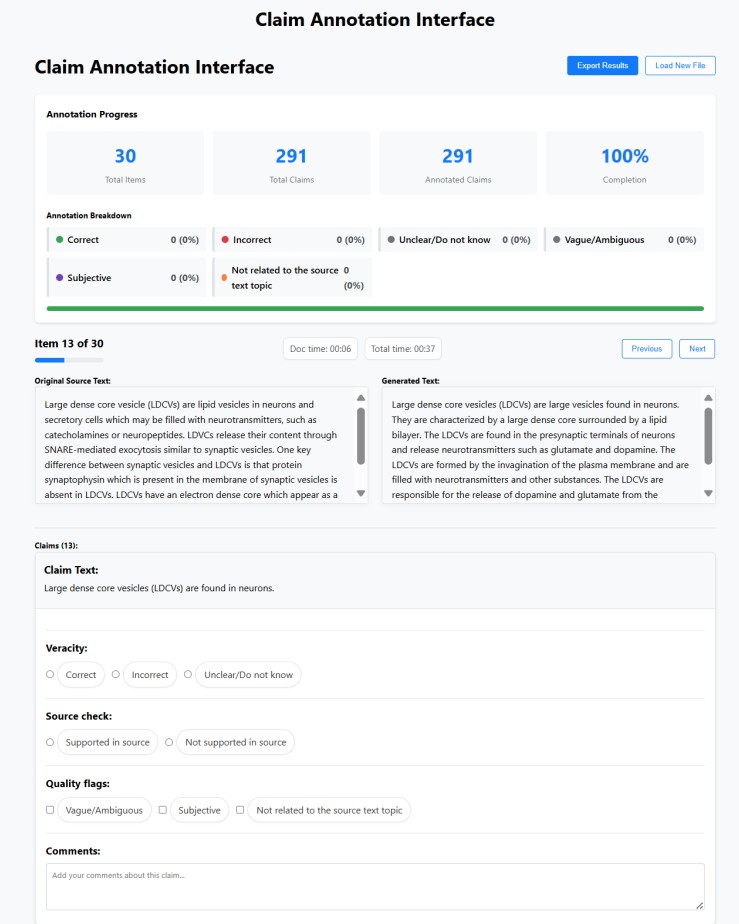

Figure 3: The interactive claim annotation tool used by participants to evaluate the factual correctness of claims from the model-generated text.

| Comparison | | DP-inf | DP-16 |
|---|---|---|---|
| Inter-Annotator | Veracity | 0.57 | 0.84 |
| | Support | 0.57 | 0.73 |
| Model vs Annotator | Support (Annotator 1) | 0.49 | 0.23 |
| | Support (Annotator 2) | 0.48 | 0.34 |

Table 10: Cohen's Kappa scores for DP-inf and DP-16 settings for human annotation of Wikipedia AI article claims.

## F FACTSCORE AND RECURRING CLAIMS ALGORITHMS

We recall the pseudo-code of FactScore in Algorithm 1 and describe our algorithm to cluster repeated claims in Algorithm 2 (cf. §3.4).

# G  FACTSCORE ALGORITHM

---

**Algorithm 1** FActScore: Atomic Fact Extraction and Verification

---

**Input:** Generated texts $\mathcal{D} = d_1, d_2...d_n$, atomic fact extractor module $\mathcal{E}$, claim verification model $\mathcal{V}$, knowledge source $\mathcal{K}$

**Output:** FactScore for each document in the generated corpus : $\mathcal{S}(\mathcal{E}, \mathcal{D})$

1: **for** each document $d_i \in \mathcal{D}$ **do**
2:     Extract a candidate set of atomic claims:

$$\mathcal{AF}_{d_i} = \mathcal{E}(d_i)$$

3:     **for** each atomic claim $\alpha_j^{(d_i)} \in \mathcal{AF}_{d_i}$ **do**
4:         Verify factuality if $\alpha_j^{(d_i)}$ is supported by knowledge source $\mathcal{K}$:

$$\hat{y}_j^{(d_i)} = \mathcal{V}(\alpha_j^{(d_i)}, \mathcal{K}) \quad \text{where } \hat{y}_j^{(d_i)} \in \{0, 1\}$$

5:     **end for**
6:     Compute per-document precision:

$$\mathcal{S}(\mathcal{E}, g) = \frac{1}{|\mathcal{AF}_{d_i}|} \sum_{j=1} \mathbb{I}(\hat{y}_j^{(d_i)} = 1)$$

7: **end for**

---

Algorithm 1 describes how the FactScore is computed for a set of generated documents. For each document $d_i$, the atomic fact extractor $\mathcal{E}$ which is an instruction-tuned LLM, is prompted with in-context examples to decompose the text into a set of atomic claims, denoted $\mathcal{AF}_{d_i}$.

Each atomic claim $\alpha_j^{(d_i)} \in \mathcal{AF}_{d_i}$ is then independently verified using an external knowledge source $\mathcal{K}$. The knowledge source is the reference data against which claims are verified, and in our setup, this consists of the relevant Wikipedia articles for the evaluation domain. For each atomic claim, evidence passages are retrieved (e.g. via BM25) from these articles which are then provided to the claim verification model. The claim verification model ($\mathcal{V}$) then judges whether the claim is supported by this knowledge, producing a binary label $\hat{y}_j^{(d_i)} \in \{0, 1\}$ that indicates whether or not the claim is factually supported.

The FactScore for the document is computed as follows:

$$\mathcal{S}(\mathcal{E}, d_i) = \frac{1}{|\mathcal{AF}_{d_i}|} \sum_j \mathbb{I}(\hat{y}_j^{(d_i)} = 1),$$

which yields a scalar score corresponding to the factual correctness of the information in the generated document with respect to the knowledge source. A higher score translates to a greater proportion of verified claims, and conversely, a lower score is indicative of fewer supported claims.

We use Llama-3.1-8B-Instruct, to perform both the atomic fact extraction as well as the atomic claim verification. To demonstrate that our results remain consistent across different choices in the claim decomposition and claim verification modules used, we also report results over other instruction-tuned LLMs such as DeepSeek-R1-Distill-Qwen-7B and Llama 3.2-3B Instruct in Table 11 and Table 12.

# H   CLUSTERING ALGORITHM DESCRIPTION

---

**Algorithm 2** Recurring Claim Cluster Algorithm

---

**Require:** Set of topics $\mathcal{T}$, where each $t \in \mathcal{T}$ has $\mathcal{S}$ corresponding generated documents about the topic. $C = \{C_t\}$: Claims per topic $t \in \mathcal{T}$, where each $C_t = \{c_i\}$ with atomic facts (supported or unsupported).

1: $\mathcal{I} \leftarrow \text{INDEXCLAIMSBYTEXT}(C, \mathcal{I})$         ▷ Index Claims to their Source Document
2: $\mathcal{K} \leftarrow \text{CLUSTERASSIGNMENTOFCLAIMS}(\mathcal{I})$     ▷ Sentence Embedding-based Agglomerative Clustering
3: $\mathcal{K} \leftarrow \text{DBSCANWITHJACCARDCLUSTERING}(\mathcal{K})$    ▷ DBSCAN Clustering over clusters to ensure their Jaccard Distance is low
4: **for** $t \in \mathcal{T}$ **do**
5:     $\mathcal{K}'[t] \leftarrow \{\}$
6: **end for**                                    ▷ Initialize $\mathcal{K}'$ to contain clusters of recurring claims
7: **for** topic $t \in \mathcal{T}$ **do**
8:     **for** Cluster $k \in \mathcal{K}(\sqcup)$ **do**
9:        **if** COUNT(S) for any $c_i \in k \geq 2$ **then**
10:           Append $k$ to $\mathcal{K}'[t]$      ▷ Append a cluster of claims if the claims contain at least two supporting documents
11:        **end if**
12:     **end for**
13: **end for**

---

# I FACTSCORE EVALUATIONS WITH DIFFERENT CLAIM DECOMPOSITION AND EVALUATION MODELS

| Temperature | Pretrained Model | Average FS | Median FS | Q1 FS | Q3 FS | Avg Max FS / Topic | Avg Min FS / Topic | # of FS >0.5 | # of FS <=0.5 |
|---|---|---|---|---|---|---|---|---|---|
| 0.3 | inf | 56.3 | 57.1 | 33.3 | 80 | 71.2 | 41.2 | 397 | 491 |
| | 16 | 54.3 | 57.1 | 28.6 | 80 | 69.7 | 38.9 | 314 | 564 |
| | 8 | 53.2 | 54.5 | 28.6 | 76.9 | 68.3 | 37.9 | 294 | 588 |
| 0.5 | inf | 55.2 | 57.7 | 33.3 | 77.8 | 68.6 | 41 | 418 | 470 |
| | 16 | 51.7 | 50 | 25 | 77.8 | 67.6 | 35.5 | 335 | 547 |
| | 8 | 52.1 | 54.2 | 28.6 | 77.1 | 67.5 | 36.7 | 332 | 548 |
| 0.7 | inf | 55 | 57.1 | 30 | 80 | 69.8 | 40.6 | 436 | 452 |
| | 16 | 52.6 | 54.2 | 28.6 | 77.8 | 67.4 | 37.9 | 380 | 502 |
| | 8 | 51.5 | 50 | 28.6 | 75 | 67.2 | 35.8 | 366 | 522 |
| 1.0 | inf | 52.6 | 53.8 | 30 | 77.8 | 67.5 | 37.8 | 442 | 446 |
| | 16 | 51.2 | 50 | 27.3 | 75 | 66.1 | 36.2 | 420 | 464 |
| | 8 | 49.8 | 50 | 25 | 72.7 | 65.4 | 34.4 | 381 | 505 |

Table 11: Factuality evaluation scores for different temperature settings for the Wikipedia Science articles, using Meta Llama 3.2-3b-Instruct for claim evaluation and decomposition.

| Temperature | Pretrained Model | Average FS | Median FS | Q1 FS | Q3 FS | Avg Max FS / Topic | Avg Min FS / Topic | # of FS >0.5 | # of FS <=0.5 |
|---|---|---|---|---|---|---|---|---|---|
| 0.3 | inf | 48.6 | 50 | 28.9 | 66.7 | 71.4 | 27.2 | 121 | 765 |
| | 16 | 47.5 | 45.5 | 25 | 66.7 | 70.9 | 24.2 | 87 | 788 |
| | 8 | 46.9 | 44.4 | 25 | 66.7 | 69 | 26 | 73 | 809 |
| 0.5 | inf | 48.9 | 50 | 30 | 66.7 | 68.7 | 29.1 | 129 | 759 |
| | 16 | 44.6 | 44.4 | 25 | 62.5 | 70.8 | 18.8 | 89 | 789 |
| | 8 | 46 | 50 | 25 | 66.7 | 65 | 27.3 | 96 | 784 |
| 0.7 | inf | 44.4 | 27.3 | 63.6 | 63.8 | 27.4 | 136 | 752 | 452 |
| | 16 | 44.4 | 27.3 | 62.5 | 68 | 23.1 | 115 | 766 | 502 |
| | 8 | 44.4 | 25 | 66.7 | 66 | 25.9 | 102 | 785 | 522 |
| 1.0 | inf | 42.9 | 25 | 62.5 | 69.2 | 22.6 | 147 | 738 | 446 |
| | 16 | 42.9 | 25 | 60 | 62.7 | 26.1 | 101 | 783 | 464 |
| | 8 | 42.9 | 27.3 | 62.5 | 63 | 25.5 | 114 | 772 | 505 |

Table 12: Factuality evaluation scores for different temperature settings for the Wikipedia Science articles, using DeepSeek-R1-Distill-Qwen-7B for claim evaluation and decomposition.

## J  FACTSCORE DISTRIBUTIONS PER TEMPERATURE SETTING UNDER DIFFERENT MODELS

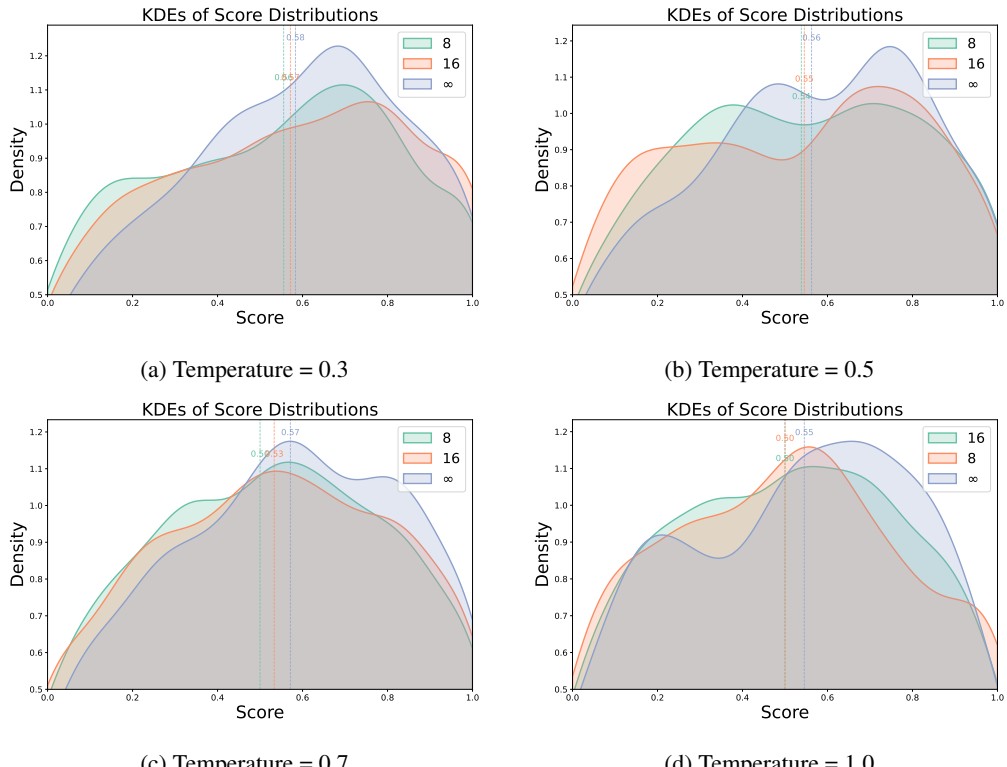

(a) Temperature = 0.3

(b) Temperature = 0.5

(c) Temperature = 0.7

(d) Temperature = 1.0

Figure 4: KDE plots of FactScore distributions of texts generated from topics in the Wikipedia Science data for under different temperature settings for GPT-J 6B.

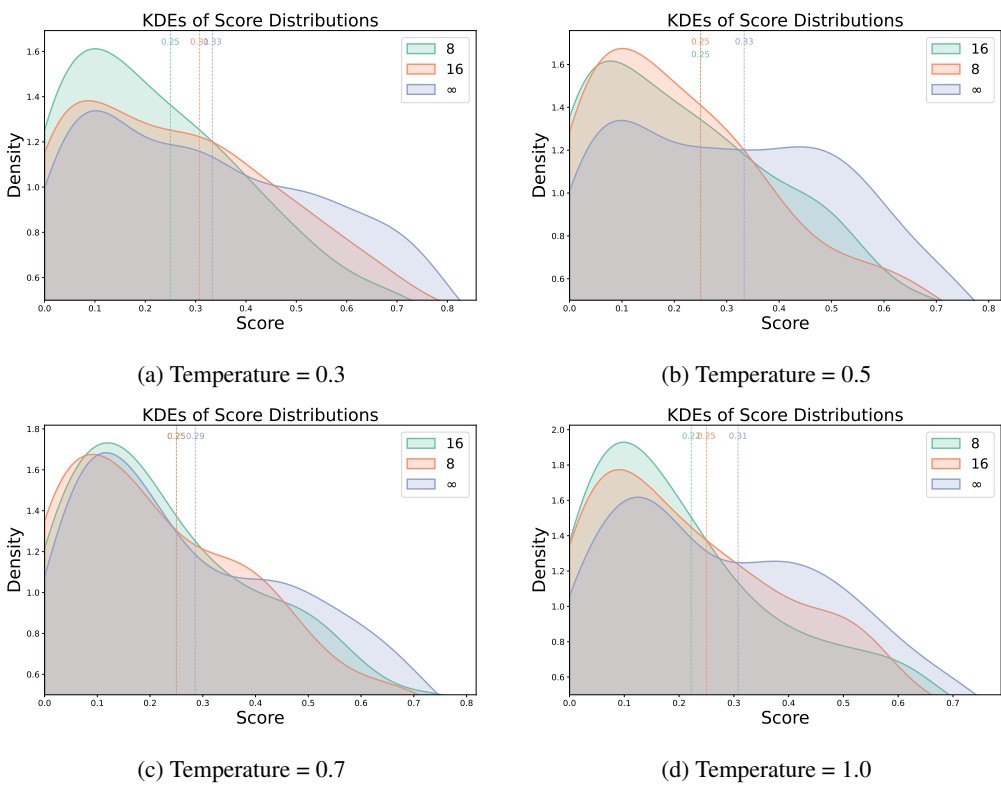

Figure 5: KDE plots of FactScore distributions of texts generated from topics in the Wikipedia AI data for under different temperature settings for GPT-J 6B.

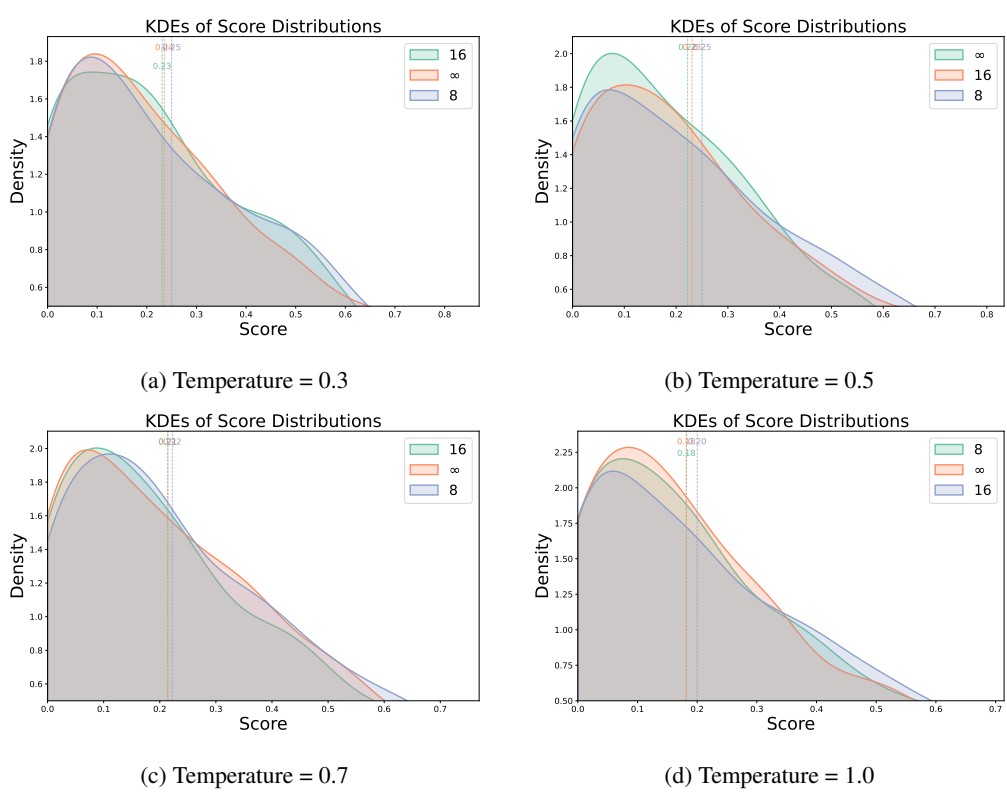

Figure 6: KDE plots of FactScore distributions of texts generated from topics in the Wikipedia Pretraining data for under different temperature settings for GPT-J 6B.

# K    FactScore Distributions per model Under Different Temperature Settings

## K.1    Models pre-trained with and without DP

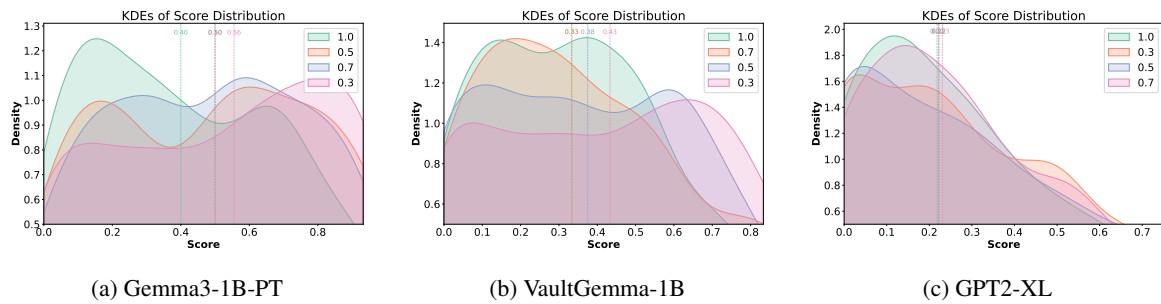

(a) Gemma3-1B-PT          (b) VaultGemma-1B          (c) GPT2-XL

Figure 7: KDE plots of FactScore distributions of texts generated from topics in the Wikipedia AI data for the pre-trained models.

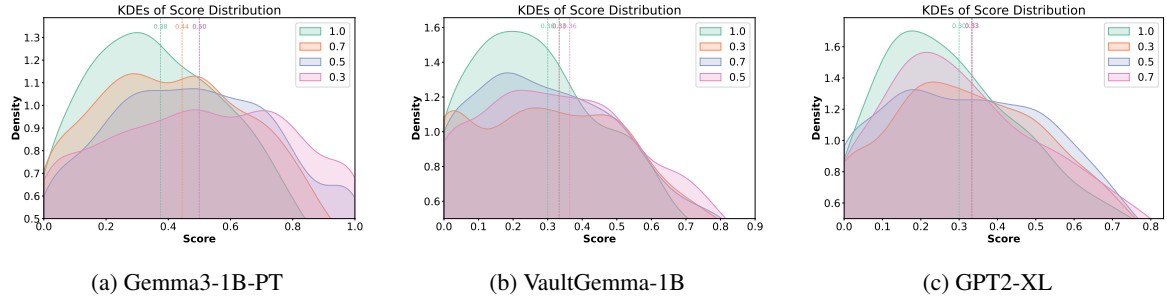

(a) Gemma3-1B-PT          (b) VaultGemma-1B          (c) GPT2-XL

Figure 8: KDE plots of FactScore distributions of texts generated from topics in the Wikipedia Science data for the pre-trained models.

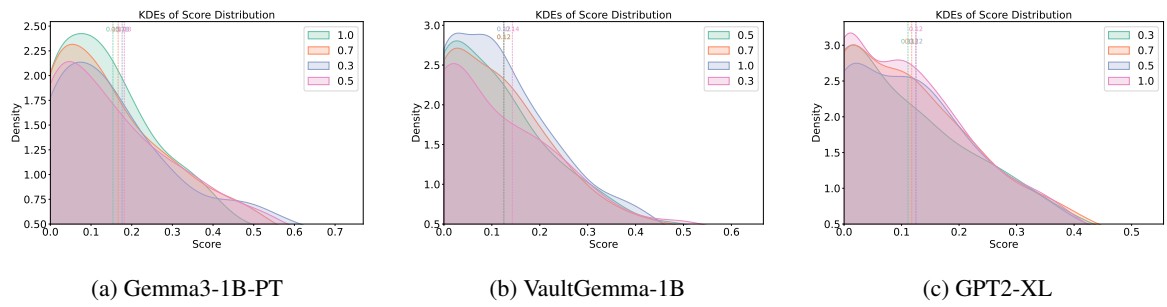

(a) Gemma3-1B-PT          (b) VaultGemma-1B          (c) GPT2-XL

Figure 9: KDE plots of FactScore distributions of texts generated from topics in the Wikipedia Pretraining data for the pre-trained models.

## K.2 MODELS FINE-TUNED WITH AND WITHOUT DP

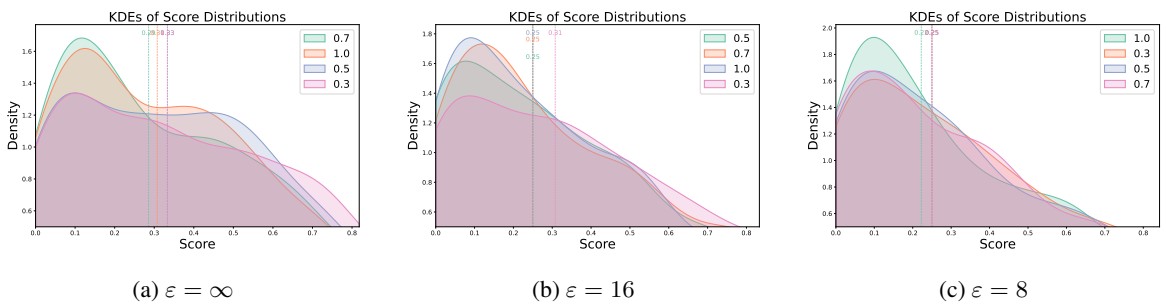

(a) $\varepsilon = \infty$      (b) $\varepsilon = 16$      (c) $\varepsilon = 8$

Figure 10: KDE plots of FactScore distributions of texts generated from topics in the Wikipedia AI data for the fine-trained models.

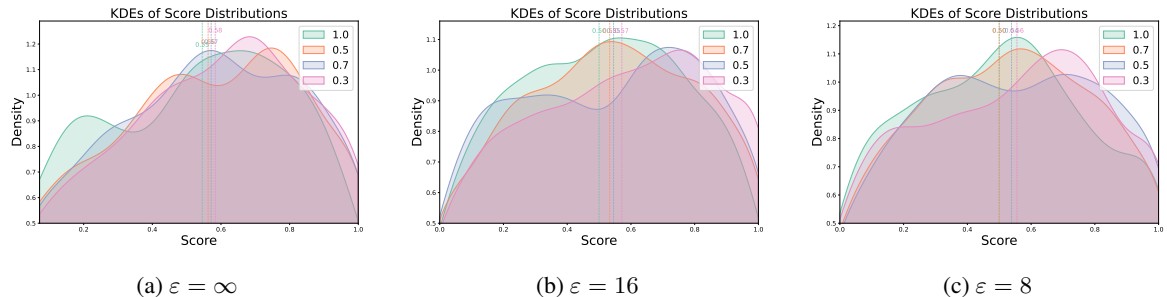

(a) $\varepsilon = \infty$      (b) $\varepsilon = 16$      (c) $\varepsilon = 8$

Figure 11: KDE plots of FactScore distributions of texts generated from topics in the Wikipedia Science data for the fine-trained models.

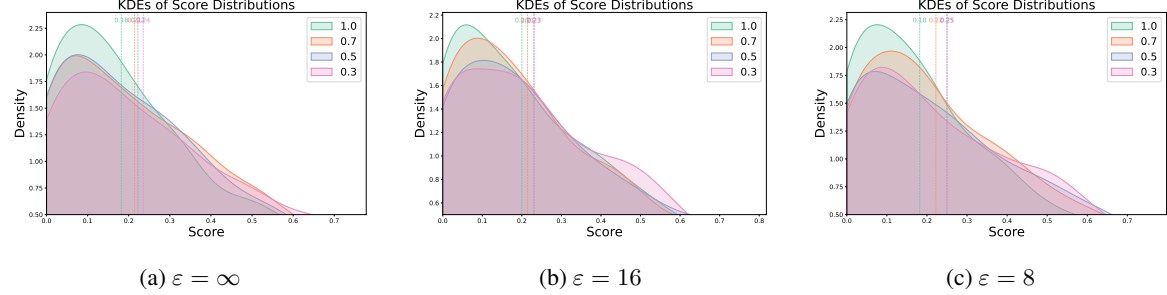

(a) $\varepsilon = \infty$      (b) $\varepsilon = 16$      (c) $\varepsilon = 8$

Figure 12: KDE plots of FactScore distributions of texts generated from topics in the Wikipedia Pretraining data for the fine-trained models.

## L    VARIABILITY IN FACTSCORE

We report the per-topic standard deviation of FactScore for Wiki AI and Wiki Science to assess the stability of model factuality across prompt variations. Figures 13a and 13b show the distributions for models trained with $\varepsilon \in \{8, 16, \infty\}$.

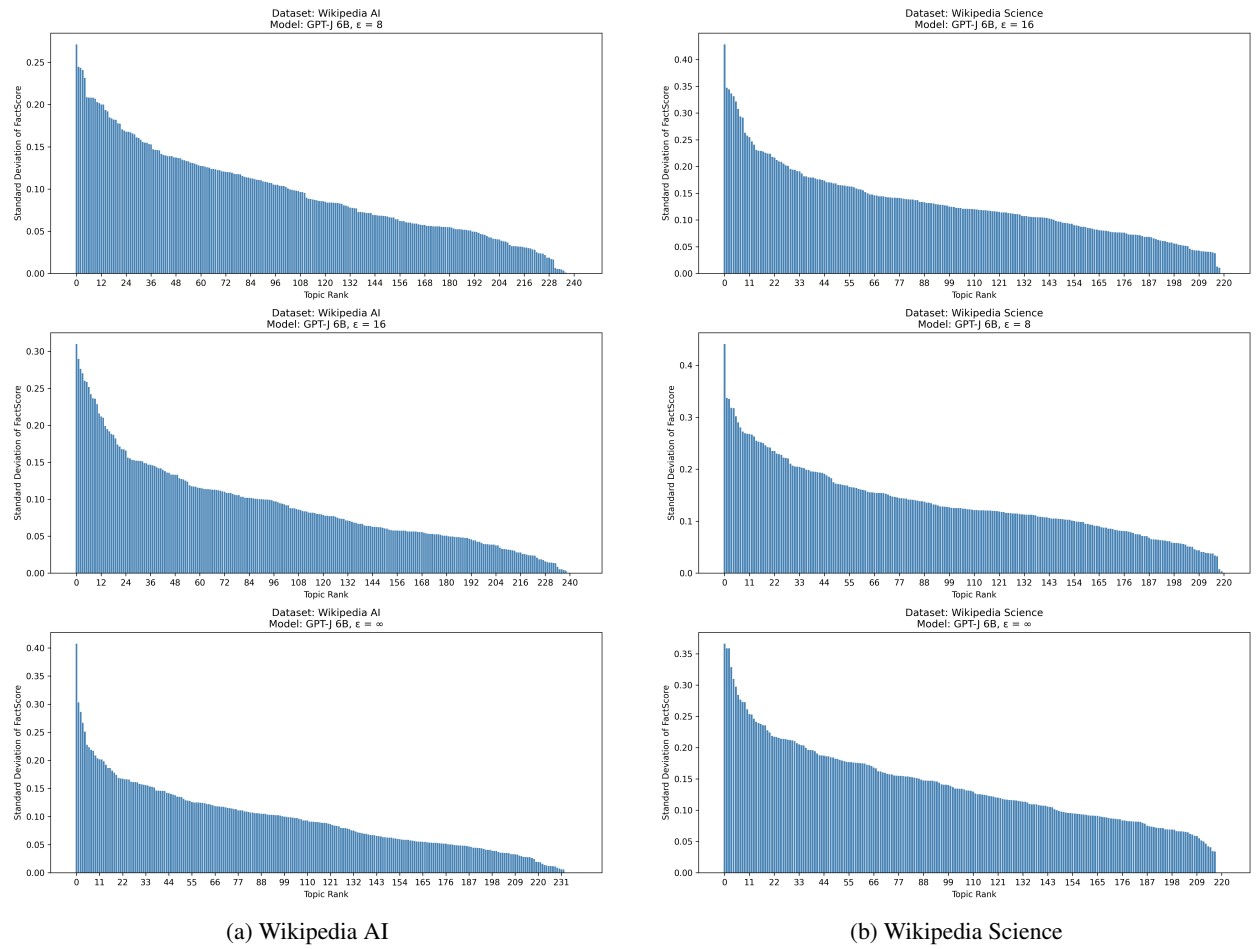

(a) Wikipedia AI                    (b) Wikipedia Science

Figure 13: Standard deviation of FactScore across topics for Wiki AI and Wiki Science.

# M  LOWER PERPLEXITY DOES NOT CORRELATE WITH FACTUAL RELIABILITY

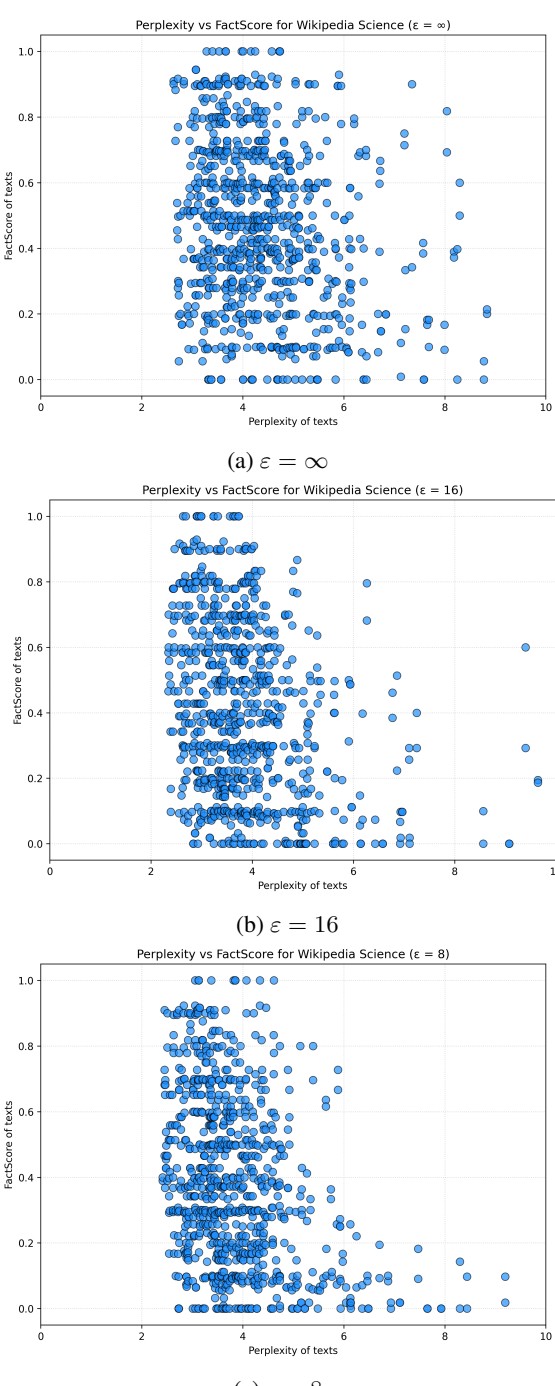

(a) $\varepsilon = \infty$

(b) $\varepsilon = 16$

(c) $\varepsilon = 8$

Figure 14: Relationship between model perplexity and FactScore for Wikipedia Science. Lower perplexity is not predictive of higher factual accuracy.

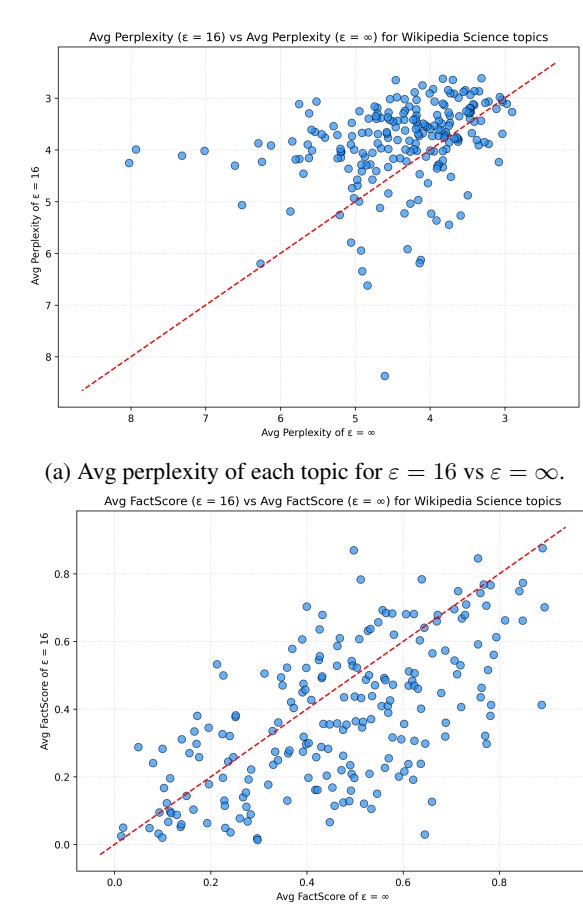

(a) Avg perplexity of each topic for $\varepsilon = 16$ vs $\varepsilon = \infty$.

(b) Avg FactScore of each topic for $\varepsilon = 16$ vs $\varepsilon = \infty$

Figure 15: Average perplexity and FactScore of topics across models. Although DP-finetuned models frequently achieve lower perplexity (points above the y = x line), the non-DP fine-tuned model attains a higher average FactScore across topics. Thus, lower perplexity does not imply higher factuality.

# N   TRAINING LOSS CURVES FOR WIKIPEDIA FINE-TUNING

We report the training loss curves for both the non-DP and DP models trained on the Wikipedia datasets. We include results from two fine-tuning settings: (1) the setup where the unseen Wikipedia articles are interspersed with pre-training Wikipedia data, and privacy is specified through a target privacy budget $\varepsilon$; and (2) fine-tuning only over the unseen Wikipedia dataset, split into 200-token articles to expand its size, where we directly set the noise multiplier due to the instability in computations of noise multipliers for large $\varepsilon$.

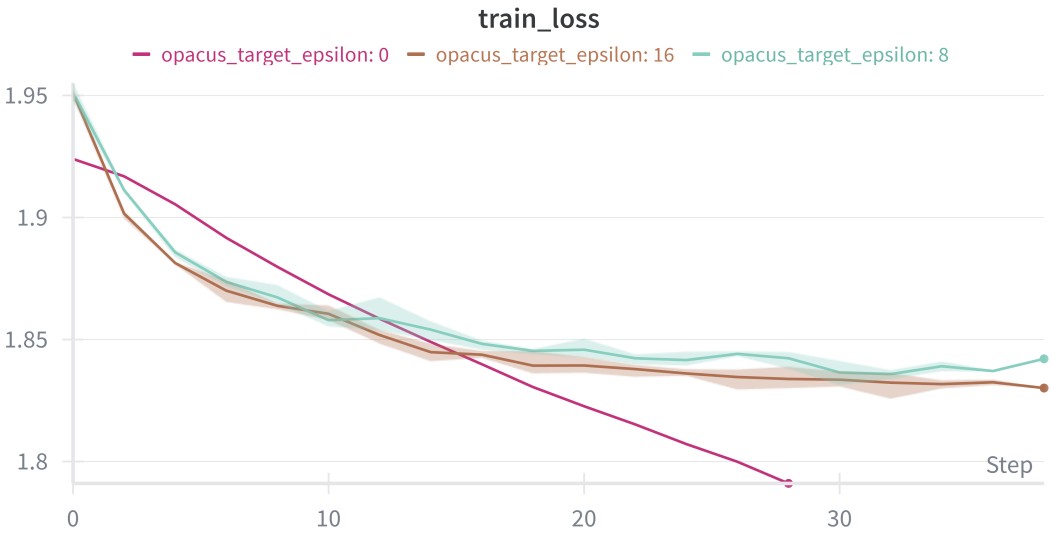

Figure 16: Training loss curve for the models fine-tuned on the large Wikipedia dataset,

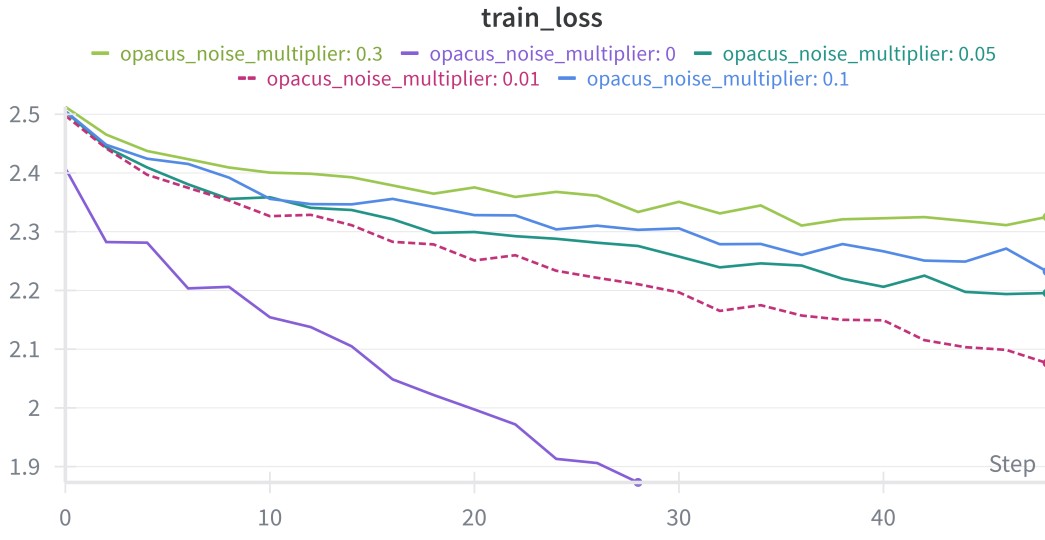

Figure 17: Training loss curve for the models fine-tuned only on the private Wikipedia articles.

## O    FACTSCORES ON UNSEEN WIKIPEDIA ARTICLES AFTER FINE-TUNING ON WIKIPEDIA DATA

| Dataset | Noise Multiplier | Average FS | Median FS | Q1 FS | Q3 FS | Avg Max FS / Topic | Avg Min FS / Topic | # of FS >0.5 | # of FS <0.5 |
|---|---|---|---|---|---|---|---|---|---|
| Wikipedia Science | inf | 24.4 | 38.5 | 20 | 55.8 | 36.7 | 12.9 | 34 | 440 |
| | 0.01 | 24.5 | 33.3 | 18.6 | 54.5 | 36.6 | 14.4 | 37 | 432 |
| | 0.1 | 22.2 | 30.4 | 11.5 | 51.3 | 35.1 | 10.1 | 41 | 427 |
| | 0.3 | 24.7 | 27.9 | 9.7 | 46.5 | 35.3 | 14.9 | 50 | 417 |
| Wikipedia AI | inf | 44.7 | 19 | 8.6 | 33.3 | 58.8 | 30.6 | 252 | 600 |
| | 0.01 | 42.1 | 18.2 | 8.3 | 35.7 | 56.7 | 27.4 | 231 | 622 |
| | 0.1 | 41.5 | 14.3 | 0 | 29.2 | 55.4 | 26.9 | 222 | 618 |
| | 0.3 | 37.3 | 17.6 | 5.9 | 35.7 | 51.5 | 23 | 175 | 680 |

Table 13: Factuality evaluation scores for temperature $\tau = 0.5$ when fine-tuning only over the unseen Wikipedia articles, using Llama-3.1-8B-Instruct for claim decomposition and verification.

| Dataset | Target Epsilon | Average FS | Median FS | Q1 FS | Q3 FS | # of FS >0.5 | # of FS <0.5 |
|---|---|---|---|---|---|---|---|
| Wikipedia Science | $\infty$ | 49 | 25.7 | 9.7 | 46.5 | 247 | 1072 |
| | 16 | 53.8 | 36.8 | 17.1 | 55.9 | 411 | 878 |
| | 8 | 52.6 | 37.2 | 18.5 | 58.4 | 427 | 860 |
| Wikipedia AI | $\infty$ | 32.7 | 18.8 | 8.6 | 34.2 | 252 | 600 |
| | 16 | 36.7 | 28.6 | 12.5 | 46 | 231 | 622 |
| | 8 | 37.4 | 29.2 | 15.4 | 46.2 | 222 | 618 |

Table 14: Factuality evaluation scores over the unseen data at temperature = 0.3 when fine-tuning only over the Wikipedia articles from pre-training, using Llama-3.1-8B-Instruct for claim decomposition and verification.

## P ADDITIONAL EXAMPLES OF CLAIM CLUSTERS

| DP Setting | Claims |
|---|---|
| **Data: Wikipedia AI, Topic: AlphaEvolve** | |
| $\varepsilon = \infty$ | 'AlphaEvolve is used for generating the molecular structures of organic molecules.', 'AlphaEvolve is used for generating molecular structures.'
'The project or system is based on the DeepChem molecular modeling framework.', 'It is based on the DeepChem molecular modeling framework.' |
| $\varepsilon = 16$ | 'AlphaEvolve is a first-person shooter.', 'AlphaEvolve is a first-person shooter video game.'
'The game features a series of missions.', 'The game features a variety of weapons.'
'There are two types of enemies in the game.', 'The game features two types of enemies.' |
| $\varepsilon = 8$ | 'Black Hole Interactive is a game development company', 'Black Hole Interactive is a video game development company.'
'The Behemoth is a studio.', 'The Behemoth is a video game development studio.', 'The Behemoth is an American studio.'
'AlphaEvolve is a shooter video game', 'AlphaEvolve is free-to-play', 'AlphaEvolve is a free-to-play video game', 'AlphaEvolve is a 2D video game.', 'AlphaEvolve is a physics-based video game.' |
| **Data: Wikipedia Science, Topic: Allogeneic processed thymus tissue** | |
| $\varepsilon = \infty$ | 'The thymus tissue is processed to remove stem cells.', ('The thymus tissue is processed to induce the recipient's immune cells to develop.'
'AML is a type of disease.', 'AML is a type of cancer.' |
| $\varepsilon = 16$ | 'CLL is also known as chronic lymphocytic leukemia.', 'CLL is often referred to as chronic lymphocytic leukemia.'
'Thymus glands are found in donor pigs.', 'The thymus glands are minced.', 'The thymus glands are from donor pigs.', 'The thymus glands are removed from pigs.' |
| $\varepsilon = 8$ | 'Dr. Daniel L. Scharff is the founder of the International Society for Cellular Therapy (ISCT).', 'Dr. Daniel L. Scharff is the former President of the International Society for Cellular Therapy (ISCT)'
'Antigen presenting cells can cause graft rejection.', 'Lymphocytes can cause graft rejection' |

Table 15: Examples of unsupported recurring claim clusters

## Q LLM USAGE

We used large language models to help with the writing of this paper. Specifically, we used ChatGPT to generate the code for LaTeX tables and figures in this research paper.

