# OpenReview forum: "The Privacy-Hallucination Tradeoff in Differentially Private Language Models"
_ICLR.cc/2026/Conference — ICLR 2026 Conference Withdrawn Submission_

### Official Review · Reviewer_49Qv · 2025-10-26

**Soundness:** 2
**Presentation:** 3
**Contribution:** 2
**Rating:** 2
**Confidence:** 4

**Summary:**

The paper studies how differential privacy (DP) affects factual accuracy in large language models, revealing a privacy-hallucination trade-off. For DP fine-tuning, the authors find that stricter privacy budgets lead to more hallucinations for facts in fine-tuning data, while pre-training knowledge remains largely unaffected. For DP pre-training, the authors find that it causes much larger factual degradation compared to non-DP pre-training. The work highlights the need for privacy-preserving methods that protect data without compromising factuality.

**Strengths:**

- The paper studies an underexplored question at the intersection of DP and factuality in LLMs

- The empirical studies are well-presented and cover both DP fine-tuning and DP pre-training

**Weaknesses:**

- Lack of conceptual novelty. The central finding is unsurprising given the well-known privacy-utility trade-off. DP inevitably reduces the model’s ability to fit the training data, and thus degrades utility, including factual accuracy. In the context of LLM fine-tuning, this manifests as poorer next-token prediction on factual tokens within the private dataset. Without analyzing overall utility (e.g., perplexity) alongside factuality, the paper's framing around a “privacy-hallucination” trade-off risks being a rebranding of the well-known privacy-utility trade-off, rather than a new phenomenon.

- The study examines only two privacy budgets (16 and 8), which are far too loose to be meaningful for most DP applications. This makes the findings difficult to interpret for practitioners concerned with realistic privacy guarantees. Moreover, the evaluation is confined to a single factuality metric (FactScore) and a narrow set of Wikipedia-style generation tasks, without testing robustness across alternative metrics, datasets, or domains. As a result, it remains unclear whether the reported privacy-hallucination effect is general or simply an artifact of this specific setup.

- The paper primarily reports an empirical observation that stricter privacy budgets increase hallucination rates, without offering a principled explanation or proposing methods to mitigate this trade-off. As a result, the work feels more like a measurement study, and does not meet the level of technical depth typically expected for ICLR.

**Questions:**

I don't have further questions.

---

> ### Author Response · Authors · 2025-12-03
> **Response to Reviewer 49Qv (1)**
>
> - In the updated PDF, we report perplexity scores for each model, in comparison to FactScores (Figure 14; Appendix M). These results show that our analysis of hallucinations targets a construct that is distinct from general notions of utility:  Figure 14 demonstrates there is no correlation between perplexity and factual correctness; several generations with low perplexity receive low factscores, and vice versa. The DP fine-tuned models often produce text with lower perplexity on average, further underscoring that other utility metrics do not necessarily capture hallucinations. A model can generate highly fluent text (low perplexity) and also hallucinate. We have also included training loss curves in Appendix N, which demonstrates that the models we evaluate do learn to fit the training data.
>
> Furthermore, as noted in the response to Reviewer 49Qv25, regardless of correlation with perplexity or other utility metrics, hallucinations represent a distinct and critical phenomenon in AI systems that warrants independent study, as evidenced by the numerous research papers focused specifically on hallucinations. For example, a single hallucinated citation in scientific research can seriously undermine credibility and violate ethical standards, irrespective of the overall utility of the model.
> - In deep learning settings, a privacy budget of $\varepsilon \approx 10$ is standard, and in general, any fixed privacy budget is considered an improvement over no enforced privacy [1]. For example, [2, 4, 5, 6] use budgets of 3, 8 and 16. Furthermore, we already find increased hallucinations at this level, suggesting the problem would only increase with more restrictive budgets.
>
> As discussed in Section 3, we put extensive thought into designing an evaluation setup where results would be valid: we focus on Wikipedia because automatic fact-checking has been best established in this domain, and we use FactScore because it is the only established method for detecting hallucinations in free-form generation, whereas QA and probing tasks conflate factual accuracy with more general utility, despite evidence that models perform well by memorizing benchmark items and task patterns [13]. We further validate that results are consistent across different implementations of FactScore (Appendix I), and we conduct human evaluations of factual accuracy (Table 3).
>
> We did consider alternative factuality evaluations, such as knowledge-probing and QA tasks. However, applying these methods to autoregressive LLMs is not straightforward. Knowledge probing typically assumes cloze-style prompts, where pre-trained LMs can fill in a missing token given bidirectional context, but autoregressively trained LLMs are trained to sequentially predict the next token. Consequently, we see high variance and poor calibration in model completions, as they are sensitive to variables such as the prompts and the decoding strategy and hyperparameters. To briefly summarize, t across 300 probes (top-k=5, max_new_tokens=5, greedy, num_beams=100) neither the DP‑16 nor DP‑inf models produce a single exact correct completion. For the log likelihoods of the correct completion, across a dataset of 1k samples, although DP‑inf had higher probability than DP‑16 on 29.9% of samples (DP‑16 higher on 32.9%, indistinguishable on 37.2%) for the correct answer.
>
> Decoding choices, tokenization strategies, and poor likelihood calibration all prevent cloze probes from eliciting the model’s true factual knowledge. In short, standard knowledge‑probing as configured here fails to reliably reveal factual knowledge in these causal LLMs. In contrast, FactScore assesses factual correctness in open-ended text generation, and gives a more reliable picture of a model's factual behavior in realistic settings where it is used to generate text [10, 11, 12, 13].
> We would be happy to consider concrete and actionable suggestions on metrics for measuring hallucination or benchmarks where automated evaluations are possible.
> - Our work offers a rigorous study on DP and hallucinations, both timely and important topics. DP is actively being explored as a “solution” to deploying AI in domains with sensitive data, especially healthcare settings. Our findings, that DP leads to increased hallucinations, are important for informing such efforts as well as generally advancing knowledge. Similar studies conducting high-impact investigations without proposing technical solutions have previously been published in ICLR (e.g. [3, 7, 8, 9]). In revisions, we have added additional discussion of implications and potential solutions.
>
> References
>
> [1] Ponomareva, et al. "How to dp-fy ml: A practical guide to machine learning with differential privacy." JAIR (2023).
>
> [2] Li, et al. "Large Language Models Can Be Strong Differentially Private Learners." ICLR (2022).
>
> [3] Staab, et al. "Beyond Memorization: Violating Privacy via Inference with Large Language Models." ICLR (2024).

---

> > ### Author Response · Authors · 2025-12-03
> > **Response to Reviewer 49Qv (1 - Contd)**
> >
> > References (Contd.)
> >
> > [4] “Fine-Tuning Language Models with Differential Privacy through Adaptive Noise Allocation”, EMNLP Findings (2024)
> >
> > [5] “Synthetic Query Generation for Privacy-Preserving Deep Retrieval Systems using Differentially Private Language Models”, NAACL 2024
> >
> > [6] “Differentially Private Language Models for Secure Data Sharing”, EMNLP 2022
> >
> > [7] “A Synthetic Dataset for Personal Attribute Inference”, NeurIPS 2024
> >
> > [8] “Can LLMs Keep a Secret? Testing Privacy Implications of Language Models via Contextual Integrity Theory”, ICLR 2024
> >
> > [9] “Re-evaluating Open-ended Evaluation of Large Language Models”, ICLR 2025
> >
> > [10] “Give Me the Facts! A Survey on Factual Knowledge Probing in Pre-trained Language Models”. (Findings of the Association for Computational Linguistics: EMNLP 2023)
> >
> > [11] “Do We Know What LLMs Don’t Know? A Study of Consistency in Knowledge Probing”. (Findings of the Association for Computational Linguistics: EMNLP 2025)
> >
> > [12] “Evaluating Open-Domain Question Answering in the Era of Large Language Models”, ACL 2023
> >
> > [13] “LASTINGBENCH: Defend Benchmarks Against Knowledge Leakage”, EMNLP Findings, 2025

---

### Official Review · Reviewer_FaGd · 2025-10-31

**Soundness:** 3
**Presentation:** 3
**Contribution:** 3
**Rating:** 6
**Confidence:** 2

**Summary:**

This paper studies the effects of differentially private fine-tuning on factuality/hallucination.

Three sets of experiments/evaluations are performed:

- Fine-tune (using DP-SGD) GPT-J model (pre-trained on data from before 2020) on new Wikipedia articles to generate articles from title.  Then, during evaluation, the model is prompted to generate articles from articles titles from a hold-out set.
    - Using automated evaluation (FactScore), they found models fine-tuned on larger privacy budgets (or no DP/standard FT) have higher FactScore compared to those with tighter budgets, leading to the conclusion that DP impacts model hallucination.  This suggests that DP hinder model's ability to acquire and generalize new knowledge.
    - Human evaluation also shows similar trends.
- They subsequently prompted the LLM to generate from titles on Wikipedia articles from before 2020 (likely in the pre-training set), and find equal performance among models trained at all privacy budget.
    - This suggests that downstream training on new knowledge with or without DP does not affect knowledge acquired from pre-training.
- They also compare and evaluate VaultGemma, which can be viewed as a DP pre-trained variant of Gemma 3-1B, and found the former to have a lower FactScore (i.e., higher hallucination)

**Strengths:**

- The paper is well-motivated and timely, and the experiments and methodology are carefully setup (including the handling of deduplication of generated facts).
- The paper adds to a cluster of recent explorations on the interaction between DP fine-tuning, regurgitation/memorization, and downstream performance. In terms of originality, however, one may argue that hallucination is just another way of viewing/framing memorization and generalization performance.

**Weaknesses:**

- It is not obvious to the reviewer whether real issue is hallucination, or the fact that the model cannot acquire real knowledge due to the DP noise.
    - It would be better informed if the author could also index the fine-tuning set and check whether the generated fact belongs to the fine-tuning set or not, so one could check whether the ability/failure to generate fact is related to memorization (which is known to be influenced by DP) or the ability to "reason" over new knowledge.  One interesting experiment would be to compare the ability to generate multiple-hop facts by chaining together one-hop facts in the fine-tuning set.
    - Could the authors comment on the frequency at which models makes statement (factual and hallucination combined), before dedup?  And if it differs across DP/non-DP models?
    - A common remedy for hallucination is to learn to abstain; if the model has learned to abstain before/after FT, what would be the influence of DP?
- A minor note is that, although the setup compares DP-SGD at different eps settings, a recent paper (https://arxiv.org/pdf/2503.12314, and references therein) shows that different DP-SGD hyperparameters calibrated to the same eps setting can have different performance (both utility and memorization).  The reviewer is curious whether the FactScore is also sensitive to the choice of hyperparameters at the same eps; if so, should we instead report the max FactScore achieved among hyperparam settings all calibrated to the same eps instead?

**Questions:**

See above

---

> ### Author Response · Authors · 2025-12-03
> **Response to Reviewer FaGd**
>
> - Our experimental setup specifically targets hallucinations: we check the factual accuracy of claims generated by each model. Our evaluations do not directly target the source of the hallucinations, e.g. if the model acquires real knowledge but fails to produce it or if the model cannot acquire real knowledge. While future work targeting this distinction can help inform model development, it has little importance to a user or practitioner: in either case, the model is unreliable. In early experiments, we did conduct template-style probing to try to uncover acquired knowledge, but we ultimately did not pursue this direction due to well-documented unreliability of knowledge probing setups for causal models [1, 2]. Additional research on measuring model acquisition of knowledge is needed to pursue this direction.
>
> We would also like to clarify that memorization and hallucinations are not inherently the same concept: while DP is expected to inhibit memorization (e.g. exact reproduction of an individual training point), inability to exactly reproduce a training data point does not necessarily translate to increased generation of incorrect facts. On the contrary, most LLM development aims to build models that generate novel/creative factually correct content: content that is neither memorized nor hallucinated.
>
> We agree that post-training methods to direct DP models to abstain more often may be a potential mitigation for the issue our work identifies, and we have revised the discussion section to include this.
> - Thank you for this suggestion. As our setup is compute-intensive (DP finetuning requires more compute than non-private fine-tuning, and FactScore involves multiple iterations of LLM-prompting which collectively are substantial), running the full pipeline over multiple settings would require substantial compute. Instead, we follow best practices for DP hyperparameter settings [3]. This setup also better aligns with a practical setting, where a practitioner would deploy a single model.
>
> References
>
> [1] “What Matters in Memorizing and Recalling Facts? Multifaceted Benchmarks for Knowledge Probing in Language Models”. EMNLP Findings 2024
>
> [2] “Do We Know What LLMs Don’t Know? A Study of Consistency in Knowledge Probing”. EMNLP Findings 2025
>
> [3] Ponomareva, et al. "How to dp-fy ml: A practical guide to machine learning with differential privacy." JAIR (2023)

---

### Official Review · Reviewer_yCFj · 2025-11-01

**Soundness:** 2
**Presentation:** 3
**Contribution:** 1
**Rating:** 2
**Confidence:** 3

**Summary:**

This paper studies the interplay between differentially private (DP) training of LLMs and the frequency of hallucinations (producing statements contradictory to facts present in the training data).
The authors consider both DP fine-tuning and DP pre-training, investigating the increase of hallucinations separately with respect to the fine-tuning or pre-training data.

The private data is chosen to be Wikipedia articles created after 2020, when the pre-training data ends. The privacy unit is chunks of 1024 tokens. The articles are either Science, where the information may appear in other forms in the pre-training data, or AI, where the information does not exist before 2020. The data is padded also with random Wikipedia articles in order to have a sufficiently large dataset. The actual "private" data is either 231 (Science) or 124 (AI) articles. Factual evaluation is done by asking the (privately-trained) LLM to produce an article given a title only. Factual claims are evaluated with respect to the true Wikipedia article using LLM-as-a-judge. Human evaluations were also done by showing CS PhD graduate students text produced by private and non-private LLMs and asking them to evaluate veracity. Finally, semantically repeating hallucinations that appear multiple times are identified through a heuristic clustering approach.

The results are split into three parts.

### Fine-tuning DP training, Fine-tuning hallucinations

In this setting, the LLM was fine-tuned on data absent in the pre-training data. The LLM was then asked to generate articles corresponding to the new data and evaluated through the LLM-as-a-judge FactScore (FS) as well as human evaluation. Results are reported for $\epsilon \in \{8,16,\infty\}$ as well as with no fine-tuning at all. The general trend is for the FS to decrease with $\epsilon$ by 3 to 5 percentage points comparing $\epsilon = \infty$ to $\epsilon = 8$. This general trend also appeared in the human eval. Curiously, the performance at $\epsilon=8$ is sometimes less than the model which has not been fine-tuned at all.

### Fine-tuning DP training, Pre-training hallucinations

Here, the LLM was fine-tuned as before, but the generated articles are from the pre-training data. Varying $\epsilon$ did not seem to make a large or consistent difference in FS performance in this case.

### Pre-training DP training, Pre-training hallucinations

Here, there is no fine-tuning. The authors compare 3 open source pre-trained models: Gemma, VaultGemma, and GPT-2. Gemma and GPT-2 are non-DP models where Gemma has been trained more recently than GPT-2. VaultGemma is a full DP trained version of Gemma. The models are asked to generate Wikipedia articles created after the training data cutoff of GPT-2. Across several domains, Gemma does better than VaultGemma which in turn does better than GPT-2. The differences range from less than 1 percentage point to 15 percentage points.

**Strengths:**

- This paper investigates the question: do privately trained models hallucinate more? Intuitively, this should be the case as there is a tension between memorization and privacy. However, the authors take the steps to empirically evaluate this claim in several settings.

- The three settings are interesting cases to study the tradeoff between privacy and hallucinations.

- While using somewhat loose measurements of hallucinations/factual validity (for example, using an LLM-as-a-judge), the authors use (limited) human eval and some ablations to show consistent results.

**Weaknesses:**

Two high-level comments. First, the empirical setup has some issues making it hard for me to completely buy the takeaways in the discussion/conclusion section (see specifics below). Second, and perhaps more important, it is hard for me to understand how this study departs from studying the privacy-utility tradeoff of a LLM to specifically the privacy-hallucination tradeoff (see final bullet below).

- Error bars showing variance of all the results are missing and are very important in this study. The gaps between reported FS are sometimes quite small and there is currently no way to evaluate the significance of these gaps.

- Dataset size is really very small with only several hundred articles being used as the data containing new facts. Given such few data, the guarantees of DP almost directly imply that hallucination on such data must be similar to that of the base model. If AlphaEvolve is only mentioned in say a couple articles, the model simply cannot learn what it is.

- The recurring hallucination analysis does not make it clear to me whether the private models have more recurring factual inconsistencies in particular, or just make more factual errors overall. The recurrence rates do not differentiate from many claims repeated twice or one claim repeated many times.

- The phenomena where the DP fine-tuned model performs worse than the base model is strange. There is another control test that would be interesting to run in the first research question. To separate the effect of DP dynamics from the training data, DP fine-tuning can be run on, say Wikipedia Science, but evaluted on generating articles from Wikipedia AI. This would also help separate effects from training on Wikipedia style articles in general vs. specific content.

- The solution to hallucination is not simply to know more. Rather, the desired behavior of a model is to cite sources and acknowledge what it knows and does not know. It would be very interesting to see how DP interplays with such mitigations, but that is not investigated in this paper. Rather the models are forced to produce a list of factual claims about a small dataset it has been privately trained on, and we see that cannot reproduce those factual claims. This is inherent to privacy, but I am not sure that this is really the full story of private models and hallucinations.

**Questions:**

- Why is the FS so low on non-privately fine-tuned models? Experience with LLMs would suggest that if trained directly on a Wikipedia article for a given topic, it should be able to return a correct article perhaps with some limited hallucinations.

- Some details seem to be missing in Appendix G about exactly how the different components of the FS algorithm are implemented (fact extractor module, claim verification model, knowledge source, prompts, etc.). I understand that Llama-3.1-8B-Instruct is used, but how exactly is not specified.

- I think the bolding in the first column of Table 6 is incorrect. Also, there should be bolding in Table 7.

---

> ### Author Response · Authors · 2025-12-03
> **Response to Reviewer yCFj (1)**
>
> Thank you for your thoughtful feedback.
>
> We first address your high-level comment:
>
> _"it is hard for me to understand how this study departs from studying the privacy-utility tradeoff of a LLM to specifically the privacy-hallucination tradeoff”_
>
> Hallucinations represent a distinct and critical phenomenon in AI systems that warrants independent study. Their importance is widely evidenced by the numerous research papers focused specifically on hallucinations (e.g. [1, 2, 3]). Hallucinations and general utility do not necessarily coincide: a model may achieve high overall effectiveness at specific tasks, but still harmfully output fabricated content. For example, a single hallucinated citation in scientific research can seriously undermine credibility and violate ethical standards, irrespective of the overall utility of the model.
>
> In the case of DP, while we may expect DP models to exhibit lower utility, this property does not inherently imply more frequent hallucinations. If a DP model is less capable of acquiring and producing factual content, it could output information that is generic or irrelevant, or it could output less content. Instead, we find that DP models produce more factually inaccurate information than non-DP models, and concerningly, they repeatedly output the same incorrect facts. These properties greatly hinder the usability of DP models in high-stakes domains, and they could not be captured with generic utility evaluations.
>
> References
>
> [1] Kalai, Adam Tauman, and Santosh S. Vempala. "Calibrated language models must hallucinate." Proceedings of the 56th Annual ACM Symposium on Theory of Computing. 2024.
>
> [2] Kalavasis, Alkis, Anay Mehrotra, and Grigoris Velegkas. "On the Limits of Language Generation: Trade-Offs between Hallucination and Mode-Collapse." Proceedings of the 57th Annual ACM Symposium on Theory of Computing. 2025.
>
> [3] Islam, Saad Obaid Ul, Anne Lauscher, and Goran Glavaš. "How Much Do LLMs Hallucinate across Languages? On Realistic Multilingual Estimation of LLM Hallucination." Proceedings of EMNLP 2025.
>
>
> Regarding specific concerns about the experimental setup:
>
> 1. In our original submission, we report KDE plots of FactScore (Figure 1, Figure 2, Appendix
> J, Appendix K) and quartiles (Table 2; Table 11, Appendix I) to provide visibility into FactScore distributions. We additionally report the standard deviation of FactScore across all topics (sampling 3 different generations per topic) in Figure 13 of the updated PDF. For some topics, the FactScore can fluctuate by as much as 0.25, which is why to ensure a fair comparison, we reported the mean of the highest FactScore achieved for each topic across multiple generations in all results Tables. Our approach is a more comprehensive evaluation than prior work on factuality evaluations [1, 2, 3], where factuality of LLM-generated text is evaluated under limited sampling constraints (e.g.: single generation, fixed temperatures).
> 2. Our training setup is sufficient for models to be able to learn. We demonstrate this in the updated PDF by providing the training loss curves in the Appendix N, which demonstrate that the model learns from the distribution. Although the experiments described in the main paper use Wikipedia articles interspersed with pretraining data, we also ran experiments directly on the full unseen Wikipedia dataset, splitting articles into 200-token segments to increase the size of the dataset. In this setup, we control the noise multiplier, as the privacy accountant cannot reliably compute a noise multiplier for very large privacy budgets specified by ε. We report the resulting loss curves and FactScores in Appendix N.
> Furthermore, even if DP models are unable to learn information from the fine-tuning data, this would not necessarily result in increased hallucinations: the model could output very generic content, irrelevant content, or nothing. Finally, dataset size is not a limitation in RQ3, as VaultGemma was trained on a larger dataset where you can expect facts to have been repeated many times across different sources, but we still see increased hallucinations in this setting.
> 3. In the recurring hallucination analysis, we first use clustering to group similar claims. Then, in Table 4, we report counts of supported and unsupported clusters, irrespective of cluster size (one claim repeated many times would still be counted as a single cluster in Table 4). Thus, Table 4 shows that DP models tend to output more recurring factual inconsistencies. In contrast, the overall FactScore metrics in Table 2 and human evaluations in Table 3 indicate that DP models output more factual inconsistencies overall.

---

> > ### Author Response · Authors · 2025-12-03
> > **Response to Reviewer yCFj (1 - Continued)**
> >
> > Regarding specific concerns about the experimental setup (contd.):
> >
> > 4. We thank the reviewer for their insightful suggestion on disentangling the effects of DP dynamics from the influence of Wikipedia training data. To address this concern, we conducted a stricter variant of the suggested control experiment, where we fine-tuned models only on the Wikipedia pre-training data (with and without DP). We explicitly excluded all unseen articles from the Wikipedia AI and Wikipedia Science corpora and evaluated the factuality of these fine-tuned models' generations on articles from both the Science and the AI domains.
> > Our results show that the models fine-tuned in this control experiment achieve substantially lower average and median FactScores than models fine-tuned on the unseen domain-specific data (interspersed with the pre-training data). Moreover, in this setting, the non-DP fine-tuned models perform worse than their DP pre-trained counterparts. These findings support our claim that the reduced hallucinations observed in the original set of experiments stem from access to topic-relevant data, rather than from stylistic alignment.
> > 5. We agree with the reviewer’s suggestion that explicit mechanisms to mitigate hallucinations, such as targeted post-training or RAG may be necessary to mitigate the issue identified by our work. Nevertheless, fully mitigating hallucinations remains an unsolved problem even in non-DP settings, and studying LLMs without explicit knowledge bases or source citing is also of interest in the field (e.g. [1-3]). Our findings conveying the impact of DP in these settings is essential groundwork for motivating explicit focus on hallucination reduction in DP models. We have revised the discussion section to further describe the implications of our work and possible mitigations.
> >
> > Questions:
> > 1. Our evaluation setup carefully ensures that Wikipedia Science and Wikipedia AI articles have minimal overlap with pre-training data: not only that exact article text was not included in pre-training data, but also that the article content focuses on concepts discovered or popularized post-2020. Thus we expect models (even without DP) to have poorer ability to re-create these articles than ones about well-established topics, where relevant information is more abundant in pre-training data. While “Wikipedia pre-training” does contain articles in pre-training data, randomly selected Wikipedia articles are often on quite obscure topics that are not necessarily prevalent in training data.
> > Additionally, we selected our base model GPT-J because of its early knowledge cut-off date and fully open pre-training data, rather than for its overall language generation capabilities. Thus performance of this model is not expected to be as strong as less transparent (and more recent) LLMs more commonly used for day-to-day interactions (e.g. OpenAI models).
> > 2. Thank you for your suggestion. We have made revisions to Appendix G to clarify details regarding the different components of the FactScore pipeline.
> > 3. Thank you for pointing out bolding errors in Tables 6 and 7. We have corrected them.
> >
> > References
> >
> > [1] FLAME : Factuality-Aware Alignment for Large Language Models, Advances in Neural Information Processing Systems 37 (NeurIPS 2024), Lin et al., 2024
> >
> > [2] An Analysis of Multilingual FActScore, Proceedings of the 2024 Conference on Empirical Methods in Natural Language Processing, Kim et al., 2024
> >
> > [3] Long-Form Information Alignment Evaluation Beyond Atomic Facts, Proceedings of the 2025 Conference on Empirical Methods in Natural Language Processing, Zheng et al., 2025

---

### Official Review · Reviewer_ybSd · 2025-11-01

**Soundness:** 2
**Presentation:** 2
**Contribution:** 3
**Rating:** 6
**Confidence:** 3

**Summary:**

This paper outlines a study of the impact of differentially private (DP) training on large language model (LLM) hallucination. They find that when models are fine-tuned using differential privacy, they exhibit more tendency to hallucinate when prompted with information from the fine-tuning set. In contrast, they don't observe the same levels of hallucination for information from the pre-training set. Finally, the test the effects of DP on pre-training, observing that it produces more hallucination than fine-tuning. They conclude that hallucination is another tradeoff / risk of DP training.

**Strengths:**

The problem they explore is valuable. While most prior work has examined accuracy-privacy tradeoffs, this work explores hallucination-privacy tradeoffs at multiple stages of training. This is an important consideration for the community. The analysis they provide gives some insights into how DP impacts hallucination, and may point toward future work in the area. Generally the writing is clear, and the structure of the paper is sound.

**Weaknesses:**

Some points are somewhat challenging to follow, and I have concerns about some of the claims made, which I outline below. If these can be addressed I would be open to raising my score.

1. The annotator-model agreement scores, particularly for support, and for DP-16 are very low. While it is claimed that humans generally agree with the model produced scores, some of these kappa values are low enough to almost indicate a lack of agreement. I would ask that the phrasing around this be changed to reflect this.

2. The text claims that models trained with eps=8 output "fewer recurring supported claims [...] and more recurring unsupported claims". However, table 4 indicates that for Wikipedia AI, models with eps=16 output more recurring unsupported claims, which is not indicated in the text or the bolding of the table.

3. While recurring hallucinations are a theme that is addressed in the introduction, it isn't discussed in depth in the paper. The analysis in table 4 is somewhat challenging to follow, and there is no analysis of what types of claims tend to be repeatedly hallucinated. I would like to see a better analysis of this in order to support the claim that models are hallucinating the same pieces of information.

4. While is it not strictly required, it would be nice to see some discussion of how this tradeoff might be improved, even as a small point in the discussion section.

**Questions:**

1. Do you have ideas for how to improve these tradeoffs?

2. What kinds of recurring hallucinations did you observe?

3. Can you clarify how the analysis in table 4 was done?

---

> ### Author Response · Authors · 2025-12-03
> **Response to Reviewer ybSd**
>
> Thank you for your thoughtful feedback and recognition of the importance of our work.
>
> Weaknesses:
> 1. We have revised L368 to clarify model-human agreement was lower in some cases: “While agreement between human annotators was generally high, model-human agreement was not always high (Table 11), indicating limitations of relying exclusively on automated evaluation. Regardless, overall trends are consistent between human and automated evaluations: both indicate greater hallucination in the DP model, even the model with a more generous privacy budget.”
> 2. Thank you for pointing out the bolding error in Table 4. We have corrected it and revised the paragraph starting at L372 accordingly.
> 3. Our original submission provides examples of recurring claims in Table 5. In revisions, we have added additional examples for each dataset to Appendix P. These examples demonstrate how the same hallucinations are repeated across generations. For example, the e=8 model falsely describes AlphaEvolve as a video game created by the fictional company “Black Hole Interactive” in separate generations.
> We did not identify any thematic trends in the recurring claims, which is likely as the claims originate from heterogeneous source topics, making it difficult to group them into meaningful types of claims that tend to be hallucinated. Accordingly, we have revised wording in the introduction and discussion to more clearly reflect what our analysis captures: specifically, that the same claims are repeatedly hallucinated (not that some types of hallucinations are more common than others).
> 4. We have expanded the discussion section to include possible mitigations, such as post-training to direct models to express uncertainty or refuse queries, DP-RAG paradigms, and preferring models trained on public data unless there is a clear strong benefit to DP-training that outweighs risks.
>
> Questions:
> 1. Please see #4 above
> 2. Please see #3 above
> 3. We provide pseudocode for the recurring claim analysis in Appendix H. In short, we first prompt models to generate documents, where we sample from the model using the same prompts (e.g. a Wikipedia article title) 3 times. We decompose each generation into claims and rate their accuracy using FactScore. Then, using sentence embeddings and agglomerative clustering, we group similar claims. In Table 4, we report counts of clusters that contain at least 2 claims from different documents. For unsupported claims, these metrics show the number of facts that are repeatedly hallucinated.

---

### Note · Authors · 2026-01-06

I have read and agree with the venue's withdrawal policy on behalf of myself and my co-authors.